# Characterization of the Erosion Basin Shaped by the Jet Flow of Sky-Jump Spillways

Raffaella Pellegrino  and Miguel Á. Toledo *

Departamento de Ingeniería Civil, Hidráulica, Energía and Medio Ambiente, Universidad Politécnica de Madrid, 28040 Madrid, Spain
* Correspondence: miguelangel.toledo@upm.es

**Abstract:** The sky-jump spillway is an economical solution to return water to rivers, but an unsuitable flip bucket design might jeopardize the spillway, the dam, or other appurtenant works. Characterizing in advance, during the design phase, the position, size, and shape of the erosion basin would be useful to ensure that water flow is returned to the river in a safe way. Also, it would be useful for the safety assessment throughout the exploitation phase when erosion has not yet reached its maximum extension. Here, based on experimental laboratory work, the location, size, and shape of the erosion basin are analyzed, and a procedure is given for its characterization according to the design of the sky-jump spillway.

**Keywords:** hydraulic structure; sky-jump spillway; flip bucket; erosion basin; scour hole; jet flow



## 1. Introduction

The flip bucket is a structure that allows the evacuated water from the spillway of a dam to return to the riverbed. There are different types of buckets: horizontal, inclined, curved, with or without teeth [1]. A procedure for estimating the position, size, and shape of the erosion basin based on the sky jump geometric configuration is provided here. The depth of the erosion basin must be determined previously, something that has been extensively discussed by several authors. A good number of empirical formulas are available that allow us to establish, with a variable approximation, the depth of the erosion basin; on the other hand, the position, plant extension, and shape of the erosion basin have received little attention. Most of the formulas for estimating the scour depth are empirical and come from the calibration of relationships obtained by regression from experimental data obtained under different test conditions. The first investigation of the erosion depth downstream of a free-falling jet was developed by Schoklitsch [2], followed by Veronese [3], who developed an empirical formula that has served as a model for most later authors [4]. It has the limitation that the erosion depth was obtained for a quasi-vertical free fall jet. Yildiz and Üzücek [5] proposed a modified Veronese formula to take into account the incidence angle of the jet when impacting on the body of water. In addition to empirical formulas, physical models [6] or numerical methods [7–9] have been used. Neural network models have also been used [10,11] as an alternative to classical regression models. Some of the proposed formulas are simple and consider only two explanatory variables: the unit flow rate and the total head, as used by Veronese [3] and Damle et al. [12]; other formulas are more complex, such as those of Rubistein [13], Yuditsky [14], Mirskhulava [15], and Zvorykin [16], which contain parameters that are difficult to measure, such as the speed of the water flow before the impact, the thickness of the jet, the turbulence coefficient, and the depth of water flow at the lowest point of the sky jump. Doddiah et al. [17] consider time as an additional parameter. Another aspect that has been also taken into consideration over time is the effect of the emulsified air, as proposed by Ervine [18], Canepa and Hager [19], and later by Pagliara et al. [20]. Recent works evaluate scour by studying the dynamic

pressures that are generated in the basin due to a free falling jet [21]. Other recent works assess the scour depth by studying the dynamic pressures that are generated in the basin taking into consideration the cushion due to the water mass [22].

Most of the formulas follow the general expression proposed by Mason [23]:

$$D = k\frac{q^x H^y}{d^z} \tag{1}$$

where, $D$ is the total scour depth, measured from the surface of the tailwater depth ($h_2$); $q$ is the unit flow; $H$ is the difference in elevation between the reservoir level and the water surface downstream of the sky jump; and $d$ the characteristic diameter of the particles. $K$, $x$, $y$, $z$ are the constants to be determined. Some authors have considered additional variables, such as the flip angle ($\alpha$) [5,24]. Few authors have tried to establish the size and position of the erosion basin by carrying out laboratory tests [25,26] or by applying neural networks [27]. Pagliara et al. [25,26] studied the characteristics of the erosion basin caused by a jet flowing out of pipes of different diameters and inclinations and proposed formulas for estimating the erosion depth, the position of the deepest point, and the width of the hole, which depend on the pipe diameter, the pipe inclination, the air content, the granulometry of the eroded material, and the fretwork downstream. They proposed an experimental equation for each of these variables. Azmathullah [10] applied nonlinear regression techniques to the data obtained from hydraulic models from other previous research works. He proposed formulas that allow the estimation of the total scour depth $D$. In addition, different techniques of computation were applied for the geometric characterization of the erosion basin, such as genetic expression programming and the adaptive neuro fuzzy inference system (ANFIS) [27]. A novel optimization algorithm HHO [28] was also proposed to improve the performance of an artificial neural network to predict scour depth; however, this study does not provide a formula for directly calculating the total scour depth.

A procedure is proposed in this paper to characterize the location, extent, and shape of the potential erosion basin as a function of the geometric parameters that define a cylindrical flip bucket. The total scour depth ($D$) can be determined using one of the available formulas (Table 1), most of which are also summarized by Castillo and Carrillo [29]. These formulas depend on the scour depth ($t$), the height of the water cushion ($h_2$), the total head ($H$), the unit flow ($q$), the grain diameter passing 50% or 90% of weight ($d_{50}$, $d_{90}$), the impingement jet angle ($\theta$), the flip angle ($\alpha$), and gravity ($g$). Using the total scour depth ($D$) widely studied by other authors, the main objective of this work is to fill a gap in the geometric characterization of the erosion basin downstream of a sky-jump spillway, providing formulas for determining the location, extent, and shape of the potential erosion basin depending of the flip bucket geometry. It should be noted that the estimation of the position, size, and shape of the erosion pit corresponds to a limit scour hole. In fact, the main goal of this study is not to predict a scour hole for a particular case but to provide a limit erosion case on the safety side regardless of soil heterogeneity and non-stationary mode of operation of the spillway. For this reason, we used an easily erodible sand, which is a material with low resistance. Furthermore, the limit scour hole is also independent of the non-stationary mode of operation of the spillway because the limit scour hole corresponds to a flow that has been maintained for a necessary time until erosion stops progressing.

**Table 1.** Empirical formulas that allow determining the basin potential erosion depth.

| Authors | | Erosion Depth (m) |
|---|---|---|
| Veronese B | [3] | $t + h_2 = 1.9\, H^{0.225} q^{0.54}$ |
| Damle A | [12] | $t + h_2 = 0.652\, q^{0.5}\, H^{0.5}$ |
| Damle B | [12] | $t + h_2 = 0.543\, q^{0.5}\, H^{0.5}$ |
| Damle C | [12] | $t + h_2 = 0.362\, q^{0.5}\, H^{0.5}$ |

**Table 1.** *Cont.*

| Authors | | Erosion Depth (m) |
|---|---|---|
| INCYTH | [30] | $t + h_2 = 1.413 q^{0.5} H^{0.25}$ |
| Schoklitsch | [2] | $t + h_2 = 0.521 \frac{H^{0.2} q^{0.57}}{d_{90}^{0.32}}$ |
| Veronese A | [3] | $t + h_2 = 0.202 \frac{q^{0.54} H^{0.225}}{d_{50}^{0.42}}$ |
| Eggenberger | [31] | $t + h_2 = 1.44 \frac{q^{0.60} H^{0.50}}{d_{90}^{0.40}}$ |
| Hartung | [32] | $t + h_2 = 1.40 \frac{q^{0.64} H^{0.36}}{d_{90}^{0.32}}$ |
| Franke | [33] | $t + h_2 = 1.13 \frac{q^{0.67} H^{0.50}}{d_{90}^{0.50}}$ |
| Mikhalev | [23] | $t + h_2 = \frac{1.804 \, q \, \sin\theta}{1 - 0.215 \cot\theta} \left( \frac{1}{d_{90}^{0.33} h_2^{0.5}} - \frac{1.126}{H} \right)$ |
| Mirtskhulava | [15] | $t + h_2 = \left( \frac{0.97}{\sqrt{d_{90}}} - \frac{1.35}{\sqrt{H}} \right) \frac{q \sin\theta}{1 - 0.175 \, \cot\theta} + 0.25 \, h_2$ |
| Chee and Kung | [24] | $t + h_2 = 3.30 \, H \left( \frac{q^2}{g H^3} \right)^{0.3} \left( \frac{H}{d} \right)^{0.1} \alpha^{0.1}$ |
| Yildiz and Üzücek | [5] | $t + h_2 = 1.9 H^{0.225} q^{0.54} \sin\theta$ |
| Martins A | [34] | $t + h_2 = 0.14 \, N - 0.73 \frac{h_2^2}{N} + 1.7 h_2$ |
| Chian Min Wu | [23] | $t + h_2 = 1.18 H^{0.235} q^{0.51}$ |
| Martins B | [35] | $t + h_2 = 1.5 \, H^{0.1} q^{0.6}$ |
| Taraimovich | [36] | $t + h_2 = 0.633 H^{0.25} q^{0.67}$ |
| Machado B | [37] | $t + h_2 = 2.98 q^{0.5} H^{0.25}$ |
| SOFRELEC | [38] | $t + h_2 = 2.3 q^{0.6} H^{0.1}$ |
| Kotoulas | [39] | $t + h_2 = 0.78 \frac{q^{0.7} H^{0.35}}{d_{90}^{0.4}}$ |
| Chee and Padyar | [40] | $t + h_2 = 2.126 \frac{q^{0.67} H^{0.18}}{d_{50}^{0.063}}$ |
| Bisaz and Tschopp | [41] | $t + h_2 = 2.76 \, q^{0.5} H^{0.25} - 7.22 \, d_{90}$ |
| Chee and Kung | [24] | $t + h_2 = 1.663 \frac{q^{0.60} H^{0.20}}{d_{50}^{0.10}}$ |
| Machado A | [37] | $t + h_2 = 1.35 \frac{q^{0.5} H^{0.3145}}{d_{90}^{0.0645}}$ |
| Jaeger | [42] | $t + h_2 = 0.6 \, H^{0.25} q^{0.5} \left( \frac{h_2}{d_{50}} \right)^{0.333}$ |
| Rubinstein | [13] | $t + h_2 = h_2 + 2.59 h_2 \left( \frac{H + h_2}{d_{90}} \right)^{0.75} \frac{q^{1.20}}{13.70 \, H^{1.80}} \left( \frac{H}{h_2} \right)^{1.33}$ |
| Mason and Arumugam | [43] | $t + h_2 = 3.27 \frac{q^{0.60} H^{0.05} h_2^{0.15}}{g^{0.30} d_{50}^{0.10}}$ |
| Ghodsian et al. | [44] | $t + h_2 = 0.75 h_2 \left( \frac{q}{(h_2^3 g)^{0.5}} \right)^{0.524} \left( \frac{d_{50}}{h_2} \right)^{-0.366} \left( \frac{H}{h_2} \right)^{0.255}$ |
| | | $N = 0.007 \sqrt[7]{\frac{Q^3 H^{1.5}}{d_{90}^2}}$ |

## 2. Materials and Methods

### 2.1. General Approach

This research was developed in four phases (Figure 1). The first phase is analysis, allowed to identify the relevant parameters for the objective pursued. In the second phase, physical modeling was developed in four steps. Initially, the experimental test campaign was defined. This was followed by the design and construction of the test facility, with all the necessary instrumentation. After that, the test protocol was defined to finally perform the tests foreseen in the programming step. From the obtained data through physical modeling, the third phase consisted of the calibration and subsequent validation of the empirical formulas that allow characterizing the erosion hole based on the geometric characteristics of the flip bucket. Finally, in the fourth and final phase, a procedure was defined and validated to characterize the erosion hole based on the geometric characteristics of the flip bucket of the spillway.

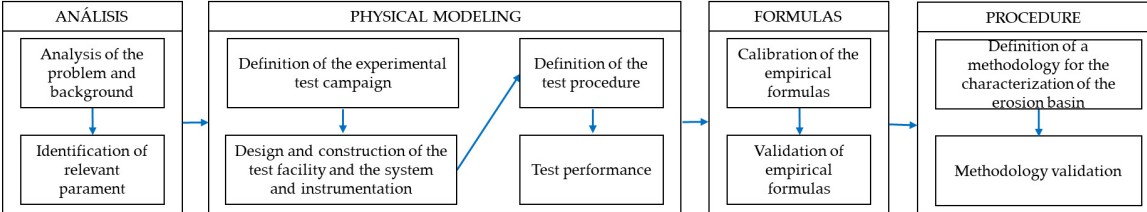

**Figure 1.** Research phases.

*2.2. Relevant Parameters*

The unit flow ($q$) and the difference between the water level at the reservoir and the water level downstream of the dam ($H$) are the parameters that most influence the depth of scour [45]. As previously commented, there are formulas that, in addition to the parameters $q$ and $H$, consider the tailwater depth ($h_2$), or the equivalent diameter of the eroded material ($d$). To study how the geometry of the flip bucket can affect the formation of the erosion basin, in addition to $q$, $H$ and $h_2$, two other parameters are defined: the radius ($R$) and the flip angle ($\alpha$) of the flip bucket. The radius of a flip bucket is usually defined in relation to the $h_b$, the water height at the launching point: $R \geq 4\,h_b$ or $R \geq 5\,h_b$, respectively, according to Peterka [46] and USBR [4]. The flip bucket usually varies between 15° and 40° [46–48]. Although there are numerous exceptions, it is unusual for the flip angle to exceed 45°. The flip angle, together with the jet speed, influences the distance where the impact occurs in the river bed or in the erosion basin. Anyway, this distance also depends, in addition to total height ($H$) and the tailwater depth ($h_2$), on the height of the lip of the flip bucket over the ground ($z_p$). Finally, the considered parameters are flow discharge ($Q$), total height ($H$), distance between the lip of the flip bucket and the level at the reservoir ($z_0$), flip bucket radius ($R$), launching angle ($\alpha$), height of the flip bucket lip over the ground ($z_p$), height of the lip of the flip bucket over the tailwater ($z_1$), tailwater depth ($h_2$), scour depth ($t$), total scour depth ($D$), distance of maximum scour from the bucket lip ($L_c$) (Figure 2), semi-axis length ($A_1$) in the direction of flow from the maximum scour point to the furthest downstream point, semi-axis length ($A_2$) in the direction of flow from the maximum scour point towards the furthest point upstream, total length ($A$) in the river longitudinal direction, semi-axis length ($B_1$) in the transverse direction to the flow from the maximum scour point towards the furthest point towards the hydraulic right (whose semi-axis length $B_2$ in the transverse direction to the flow from the maximum scour point to the furthest point to the hydraulic left), total width $B$, and shape parameter $A/B$, which we will call the circularity index (Figure 3).

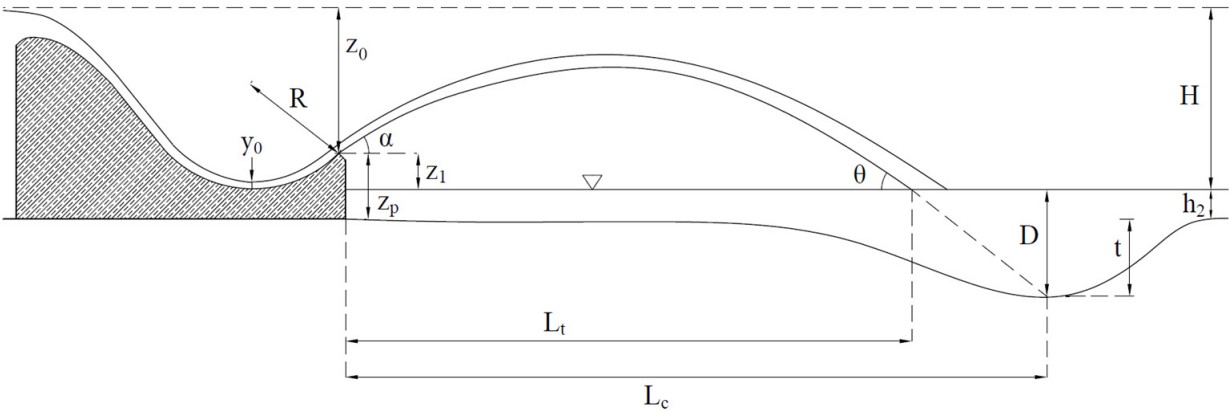

**Figure 2.** Longitudinal profile with involved parameters.

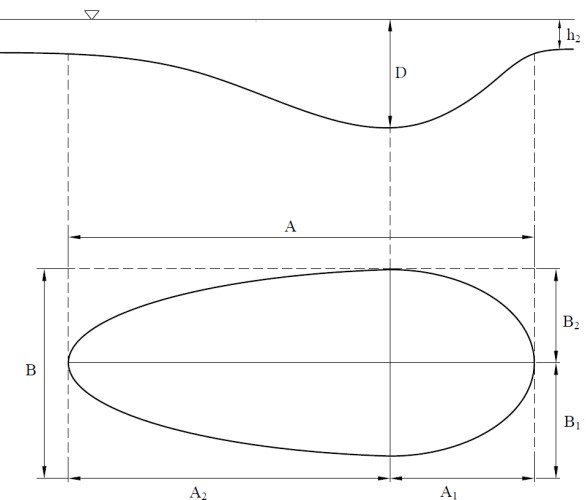

**Figure 3.** Longitudinal profile and plan of the erosion.

## 2.3. Physical Modeling

### 2.3.1. Experimental Set-Up

The tests were carried out in a channel 2.46 m wide, 1.30 m high and 13.7 m long. It is divided into three functional zones (Figure 4). The first zone is for water supply and energy dissipation. The second one is the testing area, with a length of 6.37 m, which is filled with sand up to a height of 0.50 m and is limited upstream by a wall. The sky-jump spillway is positioned in the middle of that wall. On the left side of the channel, in the direction of the flow, there is a 4.60 m long and 1.1 m high glass window, which enables visual inspection, video filming and side photography. In the third and final zone of the testing channel, there is a decanting pond to prevent any dragged material from reaching the tank, which is under the laboratory floor. Given the height limitation of the test channel and the need to have a sand thickness of 0.50 m, a height of 0.60 m was set for the spillway piece. The spillway sides are made of methacrylate (Figure 5) in order to make visible the flow of water on the sky jump. For the tests with the 0.06 m water cushion, a wooden bar sealed with silicone is positioned on top of the wall downstream of the testing area (Figure 5). For more detail on the physical model, the work of Pellegrino et al. [49] can be consulted.

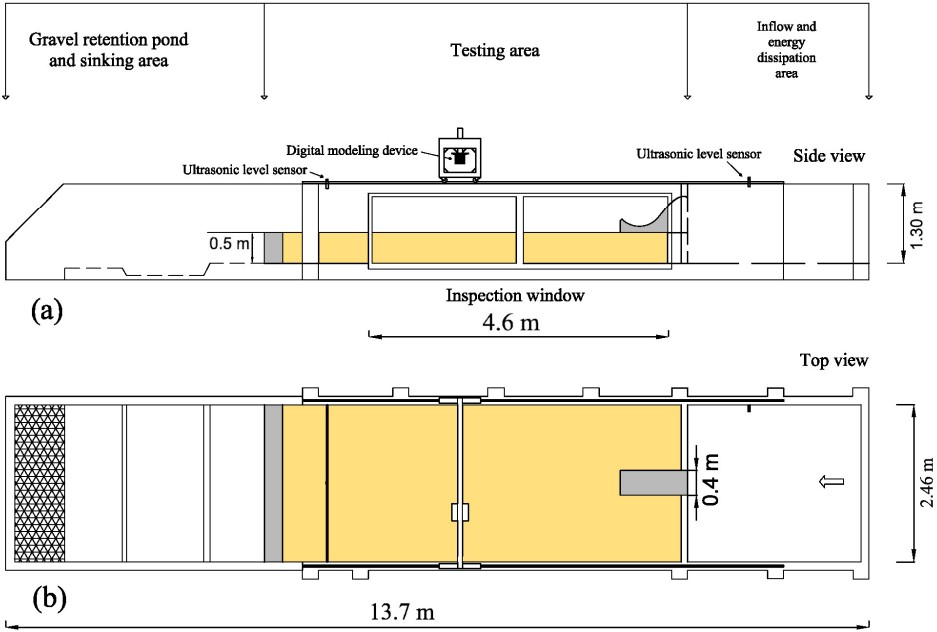

**Figure 4.** Testing set-up: (**a**) side view; (**b**) plan view.

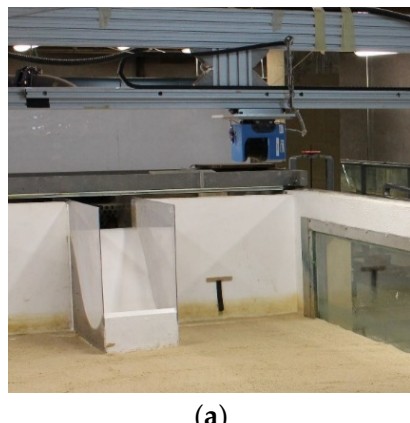
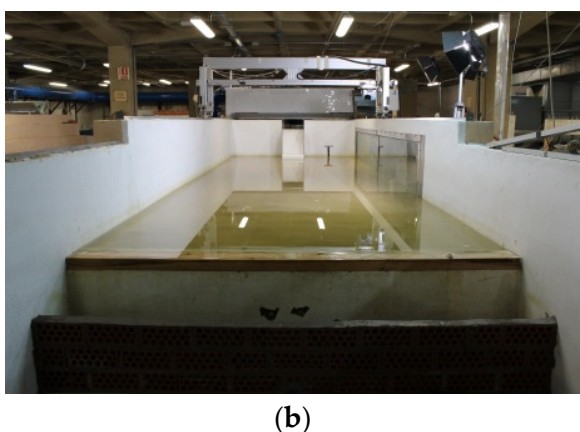

(**a**)          (**b**)

**Figure 5.** (**a**) Testing area with spillway. Case: R = 0.4 m; $\alpha$ = 45°; (**b**) view of the testing channel from downstream with a water cushion.

The facility has a 2D-laser scanner (SICK LMS200-30106, SICK AG Waldkirch, Germany) that was used to obtain the digital terrain model (DTM) before and after erosion occurred. The laser scanner is mounted on a motorized mobile structure controlled by a computer using an algorithm developed in LabView. The data obtained from the scanner are stored directly in a .txt file, and then, the DTM is obtained using the Arc-Gis tool.

As the objective of this work is to estimate the position, dimensions, and shape of the scour hole independently of the material that makes up the bed, the size of the material remained constant and small enough. Veronese [3] already observed that for a material with a size smaller than 5 mm, the erosion depth is independent of the particle size; it was later confirmed by Machado [37] and Breusers [50]. We used an easily erodible sand with a $d_{50}$ of 0.6 mm.

### 2.3.2. Test Procedure

Once the sky-jump spillway is positioned, the material is leveled and the surface of the moving bed is scanned before testing as a reference for the subsequent determination of the volume of eroded material. In the case of the tests with a water cushion 0.06 m deep, the filling with water is carried out slowly, approximately in five hours, to avoid material dragging. To verify that there was no significant settlement due to the drainage of water before scanning the erosion hole, two scans of the leveled ground were carried out, one before filling with water and another one after the slow drainage of interstitial water. From the comparison of those two DTMs, it was observed that the mean settlement was less than 1 cm. That magnitude is similar to that of the scan interval, 1 cm, that was set for scanning with the laser. Also, it is similar to the cell size used to generate the MDTs, 1 cm again. So, it can be assumed that the settlements due to the loss of interstitial water are not relevant and, consequently, that the only significant settlements are caused by the erosion process. Given the rapid erosion speed, the duration of each test was set at 12 min from the start of the spilling. The testing flow rates were obtained by opening the motorized valve for 17 s, 18 s, and 20 s, which corresponds to flow rates of 37.5 L/s, 42 L/s and 50 L/s, respectively. After each test, we waited 24 h to completely drain the canal, including the interstitial water, before scanning the surface of the eroded sandy bed (Figure 6).

The transversal profiles were obtained by means of the laser scanner every centimeter along the 2.46 m of total width. Data were stored in a text format file where the *x*, and *z* coordinates are specified for each point. Using this file and the Arc-Gis 10.3.1 software, the digital terrain model (DTM) of the scour hole was obtained for every test. To determine the maximum scour depth (*t*), the heights of the DTM obtained before and after the erosion were subtracted at every point of the mesh.

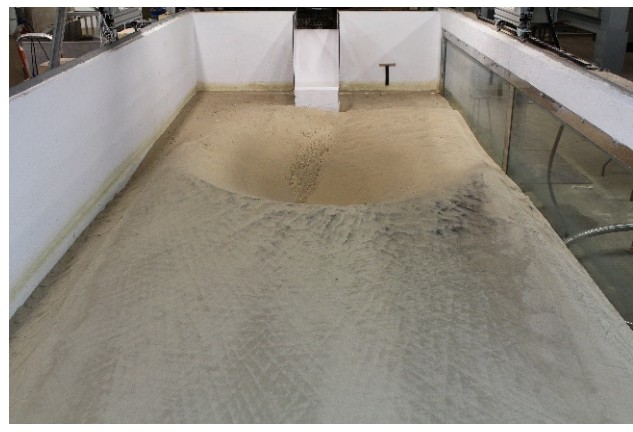
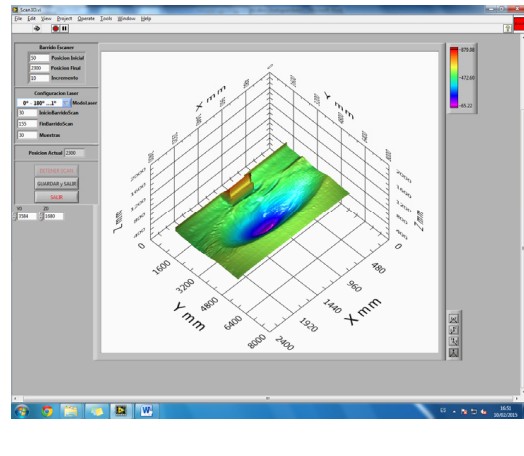

(**a**)                                                                  (**b**)

**Figure 6.** Scour hole: (**a**) photo from downstream; (**b**) output of the laser scanner.

### 2.3.3. Test Program

In order to cover the usual range of variation of the parameters, nine spillways were manufactured, with different combinations of radius and flip angle. In all cases, the spillway profile is a Creager type, the spillway is prismatic and 0.40 m wide, and the slope of the discharge channel is 1:0.8, as is usual in gravity dams. A radius of 0.20 m is combined with four flip angles (0°, 15°, 30°, 45°); a second radius of 0.30 m is combined with flip angles of 15°, 30°, and 45°; and the third radius of 0.40 m is combined with flip angles of 15° and 45°. Given the height limitation of the test channel and the need to have a sand thickness of 0.50 m, a height of 0.60 m was set for the spillway piece. The initial estimation was that with the height of 0.50 m of sand bed, a maximum flow of 50 L/s might be reached without the erosion depth exceeding the thickness of the sand. For every model, three flow rates were tested: $37.5 \pm 0.5$ L/s, $42.0 \pm 0.5$ L/s and $50.0 \pm 0.5$ L/s, which were obtained by opening the valve for 17 s, 18 s and 20 s, respectively. Two situations were considered for each spillway: without a water cushion in the channel ($h_2 = 0$ m) and with a water cushion ($h_2 = 0.06$ m). A total of 54 tests were performed.

Each of them was identified with a type name X_hi_T, where the "X" refers to the sky-jump spillway (A1, A2, A3, A4, B2, B3, B4, C2, C4); the letters A, B, C define the radius of 0.20 m, 0.30 m, 0.40 m, respectively, and the numbers 1, 2, 3, and 4 define the flip angle of 0°, 15°, 30°, and 45°, respectively; "hi" indicates the initial test condition, $h_0$ without water cushion and $h_6$ with a 0.06 m water cushion; and finally, "T" defines the valve opening time (17 s, 18 s, 20 s) (Table 2). For example, A1_h6_18 is the test with a sky jump with $R$ of 0.2 m, with $\alpha$ of 0°, with a water cushion of 0.06 m, and valve opening time of 18 s, corresponding to an experimental flow Q of 42.00 L/s.

**Table 2.** Test program.

| Flip Bucket | A1 | A2 | A3 | A4 | B2 | B3 | B4 | C2 | C4 |
|---|---|---|---|---|---|---|---|---|---|
| $R$ (m) | 0.20 | 0.20 | 0.20 | 0.20 | 0.30 | 0.30 | 0.30 | 0.40 | 0.40 |
| $\alpha$ (°) | 0 | 15 | 30 | 45 | 15 | 30 | 45 | 15 | 45 |
| $h_2$ (m) | 0.00 | 0.00 | 0.00 | 0.00 | 0.00 | 0.00 | 0.00 | 0.00 | 0.00 |
| | 0.06 | 0.06 | 0.06 | 0.06 | 0.06 | 0.06 | 0.06 | 0.06 | 0.06 |
| $Q$ (L/s) | 37.50 | 37.50 | 37.50 | 37.50 | 37.50 | 37.50 | 37.50 | 37.50 | 37.50 |
| | 42.00 | 42.00 | 42.00 | 42.00 | 42.00 | 42.00 | 42.00 | 42.00 | 42.00 |
| | 50.00 | 50.00 | 50.00 | 50.00 | 50.00 | 50.00 | 50.00 | 50.00 | 50.00 |



## 3. Results and Discussion

In the present work, experimental formulas are provided that characterize the erosion hole: position of the maximum erosion point ($L_c$), maximum length ($A$), maximum width ($B$), and circularity index ($A/B$). For the estimation of maximum depth, many formulas are available from different authors. As the erosion process is a complex physical phenomenon and it depends on the nature of the bed, rock, or soil, this work introduces the concept of limit erosion basin, and in fact the results are here summarized and analyzed for each parameter defining the limit scour hole position, size, and shape.

The limit scour hole is a pit that is formed in a fine-grained soil that may not be achieved in each real case. In fact, if the bed is made of rock, the resistance to erosion is greater than in a soil bed, so it can be affirmed that using sand corresponds to making an estimation on the safety side.

With the awareness that the risk of destruction of a structure due to a specific cause is generally difficult to determine, the procedure proposed in this work to estimate the position and size of the limit erosion basin provides a tool to assess in advance whether the erosion basin can affect adjacent structures or the spillway or dam itself. If results obtained by applying the proposed procedure indicate that the erosion does not affect the mentioned structures, it can be deduced that there will be no affectation. On the other hand, if it affects the mentioned structures, there is no certainty that the condition will occur, and more detailed studies would be required to obtain robust conclusions because in this case it would also be necessary to consider the dimensions of the limit erosion basin depending on the flows evacuated, their duration, and the quality of bed material where the spillway jet impacts. In addition, this procedure can be useful for the design of a pre-excavated scour hole.

In the present work, experimental formulas are provided that characterize the limit erosion hole: position of the maximum erosion point ($L_c$), maximum length ($A$), maximum width ($B$), and circularity index ($A/B$). For the estimation of maximum depth, many formulas are available from different authors. The results are here summarized and analyzed for each parameter defining the scour hole position, size, and shape.

The multiple linear regression techniques were applied to estimate the geometric characteristics of the limit erosion hole. The target variables are: the position of the maximum erosion point ($L_c$), the maximum length ($A$), defined as the sum of two semi-axes ($A_1$) and ($A_2$), and the circularity index ($A/B$). The explanatory variables are: the radius of the flip bucket ($R$), the impingement angle ($\theta$), the scour depth ($t$), and the difference in height ($z_1$) between the lip of the flip bucket and the downstream water level ($h_2$).

The experimental formulas were firstly calibrated and then validated; 60% of the 54 tests performed were used to calibrate the formulas. Subsequently, the relationships obtained were validated using the remaining 40% of the tests. The validation tests were chosen randomly but ensuring that all geometric typologies were included and that, for each one of them, the two conditions were also included, with and without initial cushion of water downstream. The coefficient of determination ($R^2$), absolute mean error (MAE), and relative mean error (MRE) were determined for every formula.

### 3.1. Erosion Depth

#### 3.1.1. Estimation of Erosion Depth

If the experimental data of total scour depth (*Dexp*) are compared with data calculated with the formulas detailed in the introduction of this article, it is observed that the formulas that best approximate the experimental data vary according to the calculation of $H$.

During the laboratory test, the value of parameter $H$ varies depending on the water level downstream ($h_2$). The initial condition is the one that is established before starting the test, and the parameter $h_2$ can assume only two values, 0.00 m and 0.06 m, while the stationary condition is the one that corresponds to the instant in which the entire test channel contributes to the drainage of the flow from the spillway; in this case, the value of $h_2$ will depend on this flow. It is indicated with $h_2$ the height of the water depth cushion

in the initial condition. Then, with $h_{2s}$, the height of the water cushion is the stationary condition. For each experimental test, two values of experimental total scour depth (*Dexp*) are obtained, one relating on the initial condition and the other relating to the stationary condition. Total scour depth can also be calculated by applying the literature formulas in the two conditions indicated above. For each condition, the experimental (*Dexp*) and calculated (*Dcal*) value of total scour depth are compared using the mean absolute error (MAE) and the mean relative error (MRE) (Figure 7). In the tests without a water cushion, it was observed that the erosion developed before a height cushion $h_{2s}$ (stationary condition) was established, different from the initial value $h_2$ (initial condition). For the calculation of the MAE and MRE between the experimental and calculated data of total scour (*D*), two conditions were considered: the initial condition where $h_2$ can assume values equal to 0.00 m or 0.06 m and the stationary condition where the value of $h_{2s}$ also depends on the flow drained by the spillway (Table 3).

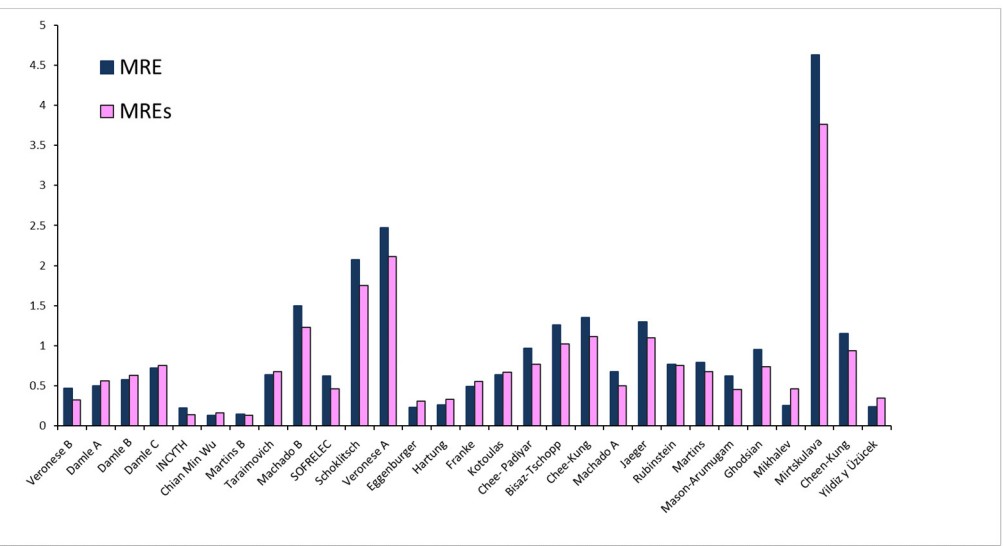

**Figure 7.** Comparing the experimental and calculated data of the total scour (D) with mean relative error in the initial condition (MRE) and in the stationary condition (MREs).

**Table 3.** Mean absolute error and mean relative error for the initial condition (MAE and MRE) and stationary condition (MAEs and MREs) obtained by comparing the experimental and calculated data of the total scour (*D*).

| Author | | MAE (m) | MRE | MAEs (m) | MREs |
|---|---|---|---|---|---|
| Veronese B | [3] | 0.159 | 0.47 | 0.116 | 0.32 |
| Damle A | [12] | 0.188 | 0.50 | 0.230 | 0.56 |
| Damle B | [12] | 0.218 | 0.58 | 0.259 | 0.63 |
| Damle C | [12] | 0.268 | 0.72 | 0.308 | 0.75 |
| INCYTH | [30] | 0.072 | 0.22 | 0.054 | 0.14 |
| Chian Min Wu | [23] | 0.050 | 0.13 | 0.069 | 0.16 |
| Martins B | [35] | 0.053 | 0.15 | 0.055 | 0.13 |
| Taraimovich | [36] | 0.238 | 0.64 | 0.277 | 0.68 |
| Machado B | [37] | 0.528 | 1.50 | 0.479 | 1.23 |
| SOFRELEC | [38] | 0.214 | 0.62 | 0.174 | 0.46 |
| Schoklitsch | [2] | 0.734 | 2.07 | 0.686 | 1.75 |
| Veronese A | [3] | 0.878 | 2.47 | 0.827 | 2.11 |
| Eggenberger | [31] | 0.090 | 0.23 | 0.132 | 0.31 |
| Hartung | [32] | 0.100 | 0.26 | 0.141 | 0.33 |
| Franke | [33] | 0.184 | 0.49 | 0.225 | 0.55 |
| Kotoulas | [39] | 0.237 | 0.64 | 0.275 | 0.67 |
| Chee-Padiyar | [40] | 0.340 | 0.97 | 0.297 | 0.77 |

**Table 3.** *Cont.*

| Author | | MAE (m) | MRE | MAEs (m) | MREs |
|---|---|---|---|---|---|
| Bisaz-Tschopp | [41] | 0.443 | 1.26 | 0.396 | 1.02 |
| Chee-Kung | [24] | 0.477 | 1.350 | 0.432 | 1.11 |
| Machado A | [37] | 0.234 | 0.68 | 0.188 | 0.50 |
| Jaeger | [42] | 0.461 | 1.299 | 0.430 | 1.099 |
| Rubinstein | [13] | 0.283 | 0.765 | 0.313 | 0.756 |
| Martins A | [34] | 0.293 | 0.791 | 0.282 | 0.680 |
| Mason-Arumugam | [43] | 0.216 | 0.620 | 0.176 | 0.451 |
| Ghodsian | [44] | 0.336 | 0.952 | 0.296 | 0.739 |
| Mikhalev | [23] | 0.087 | 0.253 | 0.185 | 0.464 |
| Mirtskulava | [15] | 1.687 | 4.628 | 1.493 | 3.758 |
| Cheen-Kung | [24] | 0.418 | 1.151 | 0.369 | 0.937 |
| Yildiz and Üzücek | [5] | 0.083 | 0.237 | 0.145 | 0.349 |

It can be observed that the formulas that best fit the experimental data are those of INCYTH [30], Chian Min Wu [23], Martins B [35], Eggenberger [31], Hartung [32], Mikhalev [23], and Yildiz and Üzücek [5] in the initial condition, while in the stationary condition the formulas of INCYTH [30], Chian Min Wu [23] and Martins B [35] have a better adjustment.

In practice, the initial condition corresponds to a poorly resistant bed; on the contrary, the stationary condition, influenced by the evacuated flow, corresponds to a resistant bed. Therefore, it is necessary to consider the type of bed to determine the scour depth. The initial condition is considered in the present study, taking into account that the sandy bed has a very low resistance to erosion.

3.1.2. Influence of the Flip Bucket Radius

The results of the tests show a moderate but clear influence of radius on the depth of erosion (Figure 8). It is observed that for the same flip angle ($\alpha$), the depth of erosion ($t$) increases when the radius ($R$) is reduced. Moreover, the effect of the radius is greater for greater angles. To quantify the influence of radium on scour, the flip angle ($\alpha$), flow rate ($Q$), and height of the tailwater ($h_2$) were fixed, and the absolute error (AE) and relative error (RE) were calculated taking the depth ($t$), marked in blue, corresponding to the greater radius ($R = 0.4$ m) as a reference (Table 4). In the case of a flip angle of 30°, where there is a lack of experimental dates, the relative error (RE) has been calculated taking the depth ($t$) corresponding to the radius equal to 0.30 m; to distinguish this situation, the results have been highlighted in yellow.

**Table 4.** Absolute errors (AR) and relative errors (RE) of the scour depth ($t$) with respect to the greater radius ($R = 0.4$ m) marked in blue for different value of flip angle ($\alpha$). The relative error, highlighted in yellow, has been calculated taking the depth ($t$) corresponding to the radius equal to 0.30 m.

| (a) ($h_2 = 0.0$ m) | | | | | | |
|---|---|---|---|---|---|---|
| Practice | Q (l/s) | $\alpha$ (°) | R (m) | t (m) | AR (m) | RE |
| A2_h0_17 | 37.5 | 15 | 0.2 | 0.321 | 0.062 | 0.24 |
| B2_h0_17 | 37.5 | 15 | 0.3 | 0.291 | 0.032 | 0.12 |
| C2_h0_17 | 37.5 | 15 | 0.4 | 0.259 | - | - |
| A2_h0_18 | 42.0 | 15 | 0.2 | 0.332 | 0.061 | 0.23 |
| B2_h0_18 | 42.0 | 15 | 0.3 | 0.310 | 0.039 | 0.14 |
| C2_h0_18 | 42.0 | 15 | 0.4 | 0.271 | - | - |
| A2_h0_20 | 50.0 | 15 | 0.2 | 0.34 | 0.055 | 0.19 |
| B2_h0_20 | 50.0 | 15 | 0.3 | 0.329 | 0.044 | 0.16 |
| C2_h0_20 | 50.0 | 15 | 0.4 | 0.285 | - | - |

**Table 4.** *Cont.*

| (b) ($h_2$ = 0.0 m) | | | | | | |
|---|---|---|---|---|---|---|
| **Practice** | **Q (l/s)** | **$\alpha$ (°)** | **R (m)** | **t (m)** | **AR (m)** | **RE** |
| A3_h0_17 | 37.5 | 30 | 0.2 | 0.416 | 0.049 | 0.13 |
| B3_h0_17 | 37.5 | 30 | 0.3 | 0.367 | - | - |
| C3_h0_17 | 37.5 | 30 | 0.4 | - | - | - |
| A3_h0_18 | 42.0 | 30 | 0.2 | 0.436 | 0.048 | 0.12 |
| B3_h0_18 | 42.0 | 30 | 0.3 | 0.388 | | - |
| C3_h0_18 | 42.0 | 30 | 0.4 | - | - | - |
| A3_h0_20 | 50.0 | 30 | 0.2 | 0.441 | 0.034 | 0.08 |
| B3_h0_20 | 50.0 | 30 | 0.3 | 0.407 | - | - |
| C3_h0_20 | 50.0 | 30 | 0.4 | - | - | - |

| (c) ($h_2$ = 0.0 m) | | | | | | |
|---|---|---|---|---|---|---|
| **Practice** | **Q (l/s)** | **$\alpha$ (°)** | **R (m)** | **t (m)** | **AR (m)** | **RE** |
| A4_h0_17 | 37.5 | 45 | 0.2 | 0.457 | 0.108 | 0.31 |
| B4_h0_17 | 37.5 | 45 | 0.3 | 0.405 | 0.056 | 0.16 |
| C4_h0_17 | 37.5 | 45 | 0.4 | 0.349 | - | - |
| A4_h0_18 | 42.0 | 45 | 0.2 | 0.475 | 0.114 | 0.32 |
| B4_h0_18 | 42.0 | 45 | 0.3 | 0.421 | 0.060 | 0.17 |
| C4_h0_18 | 42.0 | 45 | 0.4 | 0.361 | - | - |
| A4_h0_20 | 50.0 | 45 | 0.2 | 0.490 | 0.087 | 0.22 |
| B4_h0_20 | 50.0 | 45 | 0.3 | 0.443 | 0.040 | 0.10 |
| C4_h0_20 | 50.0 | 45 | 0.4 | 0.403 | - | - |

| (d) ($h_2$ = 0.06 m) | | | | | | |
|---|---|---|---|---|---|---|
| **Practice** | **Q (l/s)** | **$\alpha$ (°)** | **R (m)** | **t (m)** | **AR (m)** | **RE** |
| A2_h6_17 | 37.5 | 15 | 0.2 | 0.264 | 0.029 | 0.12 |
| B2_h6_17 | 37.5 | 15 | 0.3 | 0.253 | 0.018 | 0.08 |
| C2_h6_17 | 37.5 | 15 | 0.4 | 0.235 | - | - |
| A2_h6_18 | 42.0 | 15 | 0.2 | 0.285 | 0.035 | 0.14 |
| B2_h6_18 | 42.0 | 15 | 0.3 | 0.275 | 0.025 | 0.10 |
| C2_h6_18 | 42.0 | 15 | 0.4 | 0.250 | - | - |
| A2_h6_20 | 50.0 | 15 | 0.2 | 0.303 | 0.034 | 0.13 |
| B2_h6_20 | 50.0 | 15 | 0.3 | 0.291 | 0.022 | 0.08 |
| C2_h6_20 | 50.0 | 15 | 0.4 | 0.269 | - | - |

| (e) ($h_2$ = 0.06 m) | | | | | | |
|---|---|---|---|---|---|---|
| **Practice** | **Q (l/s)** | **$\alpha$ (°)** | **R (m)** | **t (m)** | **AR (m)** | **RE** |
| A3_h6_17 | 37.5 | 30 | 0.2 | 0.317 | 0.042 | 0.15 |
| B3_h6_17 | 37.5 | 30 | 0.3 | 0.275 | - | - |
| C3_h6_17 | 37.5 | 30 | 0.4 | - | - | - |
| A3_h6_18 | 42.0 | 30 | 0.2 | 0.320 | 0.021 | 0.07 |
| B3_h6_18 | 42.0 | 30 | 0.3 | 0.299 | | - |
| C3_h6_18 | 42.0 | 30 | 0.4 | - | - | - |
| A3_h6_20 | 50.0 | 30 | 0.2 | 0.328 | 0.01 | 0.03 |
| B3_h6_20 | 50.0 | 30 | 0.3 | 0.318 | - | - |
| C3_h6_20 | 50.0 | 30 | 0.4 | - | - | - |

**Table 4.** *Cont.*

| | | | | **(f) ($h_2$ = 0.06 m)** | | | |
|---|---|---|---|---|---|---|
| **Practice** | **Q (l/s)** | **α (°)** | **R (m)** | **t (m)** | **AR (m)** | **RE** |
| A4_h6_17 | 37.5 | 45 | 0.2 | 0.385 | 0.101 | 0.36 |
| B4_h6_17 | 37.5 | 45 | 0.3 | 0.367 | 0.083 | 0.29 |
| C4_h6_17 | 37.5 | 45 | 0.4 | 0.284 | | - |
| A4_h6_18 | 42.0 | 45 | 0.2 | 0.410 | 0.113 | 0.38 |
| B4_h6_18 | 42.0 | 45 | 0.3 | 0.380 | 0.083 | 0.28 |
| C4_h6_18 | 42.0 | 45 | 0.4 | 0.297 | - | - |
| A4_h6_20 | 50.0 | 45 | 0.2 | 0.440 | 0.122 | 0.38 |
| B4_h6_20 | 50.0 | 45 | 0.3 | 0.423 | 0.105 | 0.33 |
| C4_h6_20 | 50.0 | 45 | 0.4 | 0.318 | - | - |

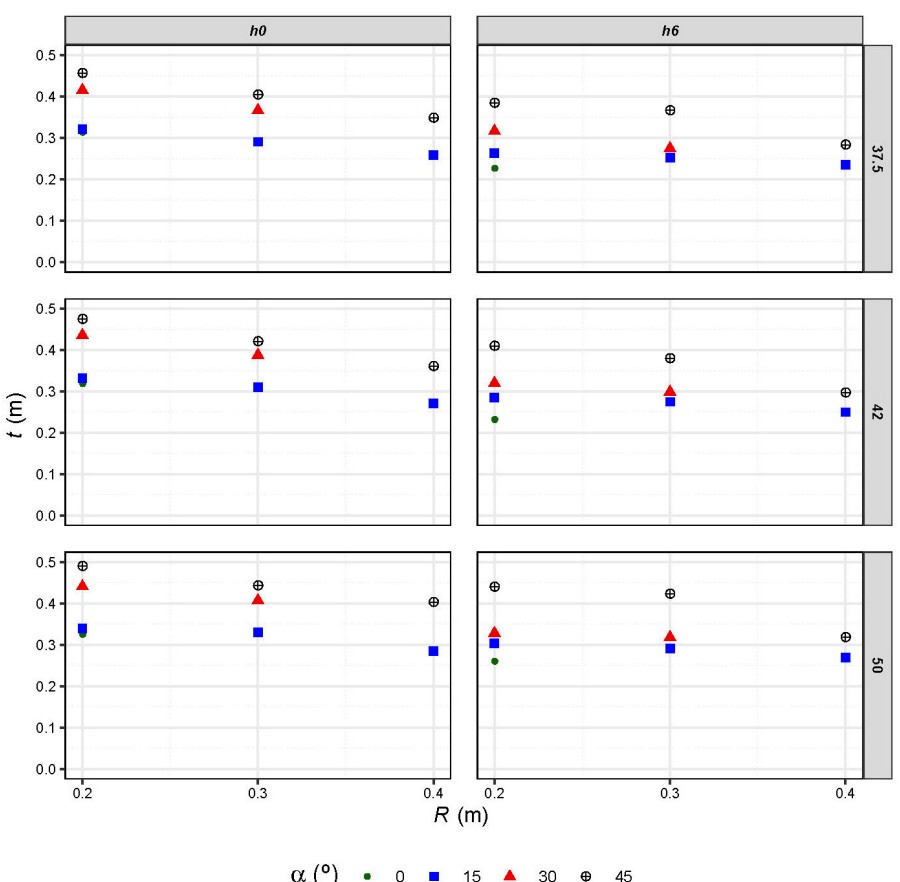

**Figure 8.** Evolution of scour depth (*t*) with the radius (*R*), separating the data, for each flip angle (*α*), by height of the water cushion ($h_0$, without cushion, and $h_6$, with cushion of 0.06 m) and by test flow rate (37.5 L/s, 42 L/s and 50 L/s).

The scour depth increases 25% on average when the radius decrease from 0.40 m to 0.20 m. This difference is significant and might be a reason to design the flip bucket with a radius greater than the minimum necessary, although this increases the flip bucket cost.

### 3.1.3. Influence of the Flip Angle

The results of the tests clearly show that erosion increases as the flip angle (*α*) becomes greater (Figure 9).

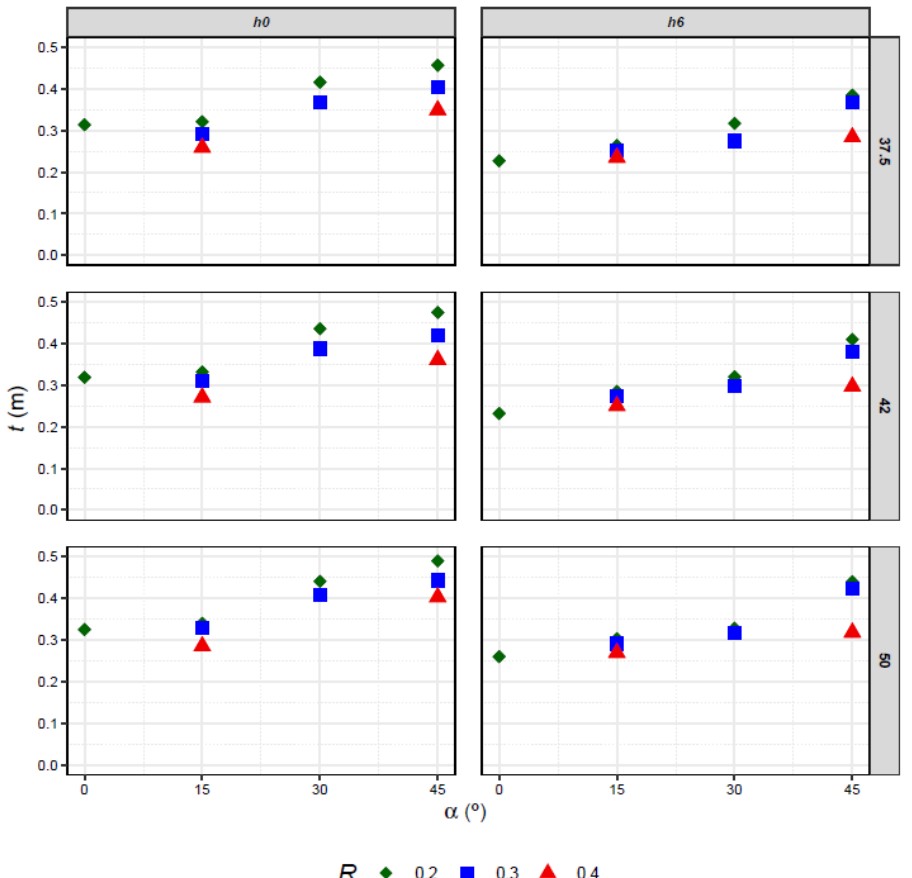

**Figure 9.** Evolution of scour depth ($t$) with the flip angle ($\alpha$), separating the data, for each radius, by height of the water cushion ($h_0$, without cushion, and $h_6$, with cushion of 0.06 m) and by test flow rate (37.5 L/s, 42 L/s and 50 L/s).

To quantify the influence of the flip angle on the scour depth ($t$), the radius ($R$), the flow rate ($Q$), and the water cushion ($h_2$) were fixed, and the absolute error (AE) and the relative error (RE) were calculated taking the scour depth ($t$) marked in blue corresponding to the greater angle ($\alpha = 45°$) as a reference (Table 5).

**Table 5.** Absolute errors (AR) and relative errors (RE) of the scour depth ($t$) with respect to the greater flip angle ($\alpha = 45°$) marked in blue, for different values of radius ($R$).

| Practice | Q (l/s) | $\alpha$ (°) | R (m) | t (m) | AR (m) | RE |
|---|---|---|---|---|---|---|
| \multicolumn{7}{c}{(g) ($h_2$ = 0.00 m)} |
| A2_h0_17 | 37.5 | 15 | 0.2 | 0.321 | 0.136 | 0.30 |
| B2_h0_17 | 37.5 | 30 | 0.2 | 0.416 | 0.041 | 0.09 |
| C2_h0_17 | 37.5 | 45 | 0.2 | 0.457 | - | - |
| A2_h0_18 | 42.0 | 15 | 0.2 | 0.332 | 0.143 | 0.30 |
| B2_h0_18 | 42.0 | 30 | 0.2 | 0.436 | 0.039 | 0.08 |
| C2_h0_18 | 42.0 | 45 | 0.2 | 0.475 | - | - |
| A2_h0_20 | 50.0 | 15 | 0.2 | 0.340 | 0.150 | 0.31 |
| B2_h0_20 | 50.0 | 30 | 0.2 | 0.441 | 0.049 | 0.10 |
| C2_h0_20 | 50.0 | 45 | 0.2 | 0.490 | - | - |

**Table 5.** *Cont.*

| (h) ($h_2$ = 0.0 m) | | | | | | |
|---|---|---|---|---|---|---|
| **Practice** | **Q (l/s)** | **α (°)** | **R (m)** | **t (m)** | **AR (m)** | **RE** |
| A3_h0_17 | 37.5 | 15 | 0.3 | 0.291 | 0.114 | 0.28 |
| B3_h0_17 | 37.5 | 30 | 0.3 | 0.367 | 0.038 | 0.09 |
| C3_h0_17 | 37.5 | 45 | 0.3 | 0.405 | - | - |
| A3_h0_18 | 42.0 | 15 | 0.3 | 0.310 | 0.111 | 0.26 |
| B3_h0_18 | 42.0 | 30 | 0.3 | 0.388 | 0.033 | 0.08 |
| C3_h0_18 | 42.0 | 45 | 0.3 | 0.421 | - | - |
| A3_h0_20 | 50.0 | 15 | 0.3 | 0.330 | 0.113 | 0.26 |
| B3_h0_20 | 50.0 | 30 | 0.3 | 0.407 | 0.036 | 0.08 |
| C3_h0_20 | 50.0 | 45 | 0.3 | 0.443 | - | - |
| (i) ($h_2$ = 0.0 m) | | | | | | |
| **Practice** | **Q (l/s)** | **α (°)** | **R (m)** | **t (m)** | **AR (m)** | **RE** |
| A4_h0_17 | 37.5 | 15 | 0.4 | 0.259 | 0.09 | 0.26 |
| B4_h0_17 | 37.5 | 30 | 0.4 | - | - | - |
| C4_h0_17 | 37.5 | 45 | 0.4 | 0.349 | - | - |
| A4_h0_18 | 42.0 | 15 | 0.4 | 0.271 | 0.09 | 0.25 |
| B4_h0_18 | 42.0 | 30 | 0.4 | - | - | - |
| C4_h0_18 | 42.0 | 45 | 0.4 | 0.361 | - | - |
| A4_h0_20 | 50.0 | 15 | 0.4 | 0.285 | 0.118 | 0.29 |
| B4_h0_20 | 50.0 | 30 | 0.4 | - | - | - |
| C4_h0_20 | 50.0 | 45 | 0.4 | 0.403 | - | - |
| (jl) ($h_2$ = 0.06 m) | | | | | | |
| **Practice** | **Q (l/s)** | **α (°)** | **R (m)** | **t (m)** | **AR (m)** | **RE** |
| A2_h6_17 | 37.5 | 15 | 0.2 | 0.264 | 0.121 | 0.31 |
| B2_h6_17 | 37.5 | 30 | 0.2 | 0.317 | 0.068 | 0.18 |
| C2_h6_17 | 37.5 | 45 | 0.2 | 0.385 | - | - |
| A2_h6_18 | 42.0 | 15 | 0.2 | 0.285 | 0.125 | 0.30 |
| B2_h6_18 | 42.0 | 30 | 0.2 | 0.320 | 0.09 | 0.22 |
| C2_h6_18 | 42.0 | 45 | 0.2 | 0.410 | - | - |
| A2_h6_20 | 50.0 | 15 | 0.2 | 0.303 | 0.137 | 0.31 |
| B2_h6_20 | 50.0 | 30 | 0.2 | 0.328 | 0.112 | 0.25 |
| C2_h6_20 | 50.0 | 45 | 0.2 | 0.440 | - | - |
| (k) ($h_2$ = 0.06 m) | | | | | | |
| **Practice** | **Q (l/s)** | **α (°)** | **R (m)** | **t (m)** | **AR (m)** | **RE** |
| A3_h6_17 | 37.5 | 15 | 0.3 | 0.253 | 0.114 | 0.31 |
| B3_h6_17 | 37.5 | 30 | 0.3 | 0.275 | 0.092 | 0.25 |
| C3_h6_17 | 37.5 | 45 | 0.3 | 0.367 | - | - |
| A3_h6_18 | 42.0 | 15 | 0.3 | 0.275 | 0.105 | 0.28 |
| B3_h6_18 | 42.0 | 30 | 0.3 | 0.299 | 0.081 | 0.21 |
| C3_h6_18 | 42.0 | 45 | 0.3 | 0.380 | | |
| A3_h6_20 | 50.0 | 15 | 0.3 | 0.291 | 0.132 | 0.31 |
| B3_h6_20 | 50.0 | 30 | 0.3 | 0.318 | 0.105 | 0.25 |
| C3_h6_20 | 50.0 | 45 | 0.3 | 0.423 | - | - |

**Table 5.** *Cont.*

| | (l) ($h_2 = 0.06$ m) | | | | | |
|---|---|---|---|---|---|---|
| **Practice** | **Q (l/s)** | **$\alpha$ (°)** | **R (m)** | **t (m)** | **AR (m)** | **RE** |
| A4_h6_17 | 37.5 | 15 | 0.4 | 0.235 | 0.049 | 0.17 |
| B4_h6_17 | 37.5 | 30 | 0.4 | - | - | - |
| C4_h6_17 | 37.5 | 45 | 0.4 | 0.284 | - | - |
| A4_h6_18 | 42.0 | 15 | 0.4 | 0.250 | 0.047 | 0.16 |
| B4_h6_18 | 42.0 | 30 | 0.4 | - | - | - |
| C4_h6_18 | 42.0 | 45 | 0.4 | 0.297 | - | - |
| A4_h6_20 | 50.0 | 15 | 0.4 | 0.269 | 0.049 | 0.15 |
| B4_h6_20 | 50.0 | 30 | 0.4 | - | - | - |
| C4_h6_20 | 50.0 | 45 | 0.4 | 0.318 | - | - |

Scour depth ($t$) increases 27% on average when angle changes from 15° to 45°. This difference is significant and might be a reason to design the flip bucket with a smaller flip angle. Furthermore, the effect of the flip angle increases when the radius is the lower value, 0.20 m.

### 3.2. Position of the Point of Maximum Erosion Depth

The parameter $L_c$ defines the distance in plan from the lip of the flip bucket to the point where erosion depth is maximum. It is observed that $L_c$ increases for values of $\alpha$ between 0° and 30°; however, it decreases for 45° (Figure 10). We should take into consideration that there are two opposite effects: (a) when the angle increases, up to 45°, the impact on the terrain surface occurs farther away from the flip bucket; (b) also, when the angle increases, the jet impacts the ground with a more vertical direction, which implies that erosion moves less downstream and develops more in the vertical direction, causing a deeper erosion. As a consequence, an increase in the flip angle might result in a greater or lower $L_c$, depending on the prevalence of one of the two mentioned effects

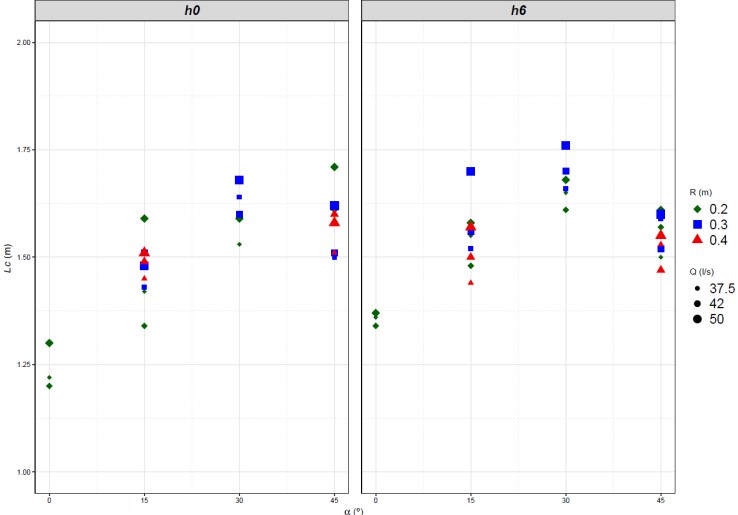

**Figure 10.** Evolution of the position of the point of maximum erosion depth $L_c$ with the flip angle ($\alpha$), separating the cases without cushion $h_0$ and with cushion $h_6$ and distinguishing by size the different flows (37.5, 42.0, 50.0 L/s) and by shape and color the radii (0.2, 0.3 and 0.4 m).

For the calculation of the position of the point with maximum erosion depth $L_c$, two linear regressions were performed: one considering the parameter $t/z_1$ and the impingement angle ($\theta$) and another considering $t/z_1$ and the radius ($R$). Calibration and validation sets were separated. The first formula (Equation (2)) presents an $R^2$ of 0.982, and MAEv equal to 0.19 m and an MREv of 12.76%, corresponding to the validation set. The second formula

(Equation (3)) presents an $R^2$ of 0.971, an MAEv equal to 0.265 m, and an MREv of 17.44%. Although the first equation is more precise, $R$ is directly available data, while $\theta$ must be determined [51].

$$\frac{L_c}{z_1} = \left(5.76\,\frac{t}{z_1} - 0.1\,\theta\right) \qquad (2)$$

$$\frac{L_c}{z_1} = \left(5.28\,\frac{t}{z_1} - 3.97\,R\right) \qquad (3)$$

Equations (2) and (3) express linear relations between $L_c/z_1$ and $\theta$ and $R$, respectively. This function was represented together with the experimental data. (Figures 11 and 12). We can observe a reasonable correspondence with the experimental values, as expected from the error quantification. Furthermore, the parameter $L_c/z_1$ decreases when the impingement angle ($\theta$) increases. As expected, the existence of a water cushion clearly affects the position of the point with maximum erosion depth. When there is a water cushion, the geometry of the flip bucket has a more significant influence on the parameter $L_c/z_1$ (Figure 13).

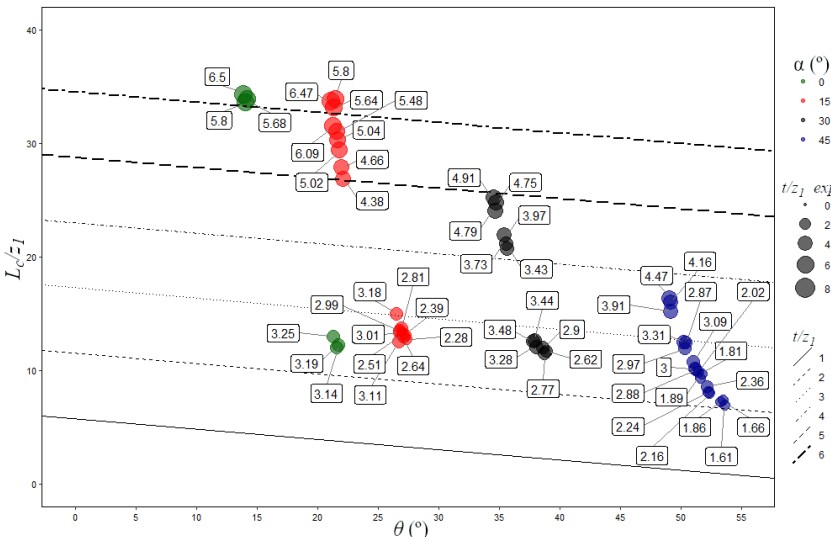

**Figure 11.** Results obtained for $L_c/z_1$ with the Equation (2) for several values of $t/z_1$ and impingement angle ($\theta$). The experimental value of $t/z_1$ is indicated next to each point.

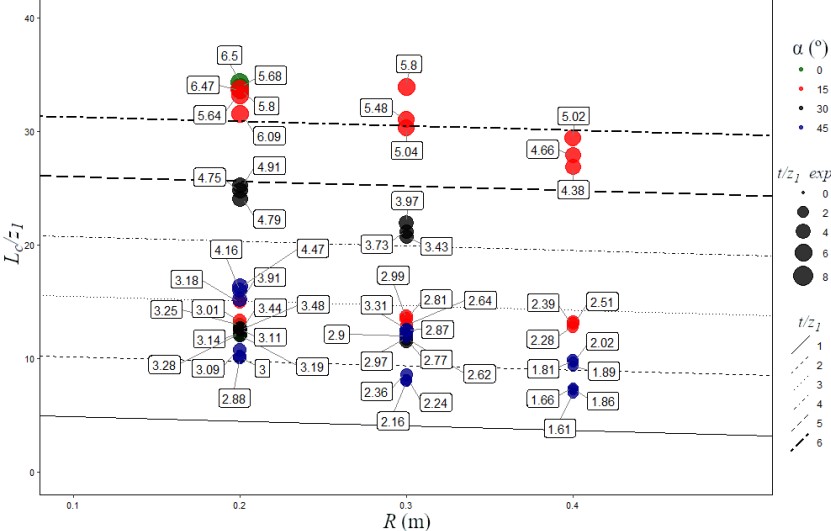

**Figure 12.** Results obtained for $L_c/z_1$ with the Equation (3) for several values of $t/z_1$ and radius ($R$). The experimental value of $t/z_1$ is indicated next to each point.

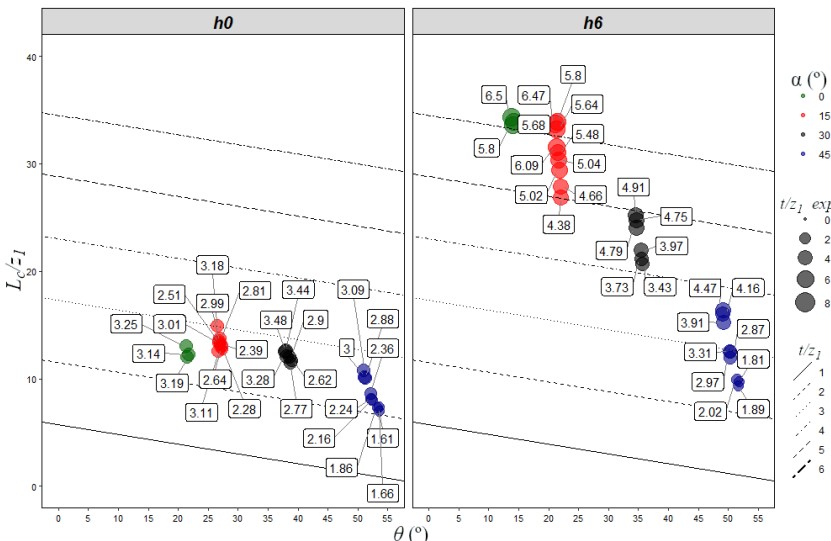

**Figure 13.** Results obtained for $L_c/z_1$ with the Equation (2) for several values of $t/z_1$ and impingement angle ($\theta$). The experimental value of $t/z_1$ is indicated next to each point (without water cushion, $h_0$, and with water cushion, $h_6$).

### 3.3. Length of the Erosion Basin

The size of the limit erosion basin was defined by parameters *A* and *B*, maximum length and width, respectively, at the level of the sandy bed, 500 mm above the bottom of the testing channel. However, the contour corresponding to the 500 mm height is sometimes open, probably due to the effect of the boundary conditions imposed by the walls of the channel and the flip bucket itself. In order to overcome this difficulty, alternative parameters *a* and *b* were considered, with the same meaning as the previous *A* and *B*, but defined at the height of 400 mm. It is the highest contour of the erosion hole that is closed in all the tests, which seems to indicate that the effect of the boundary conditions is irrelevant or null at that level (Figure 14).

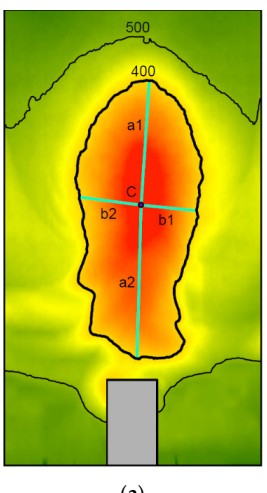

(a)

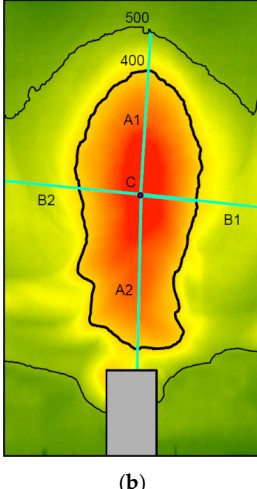

(b)

**Figure 14.** (**a**) For contour line at height of 400 mm is the limit scour hole semi-axis from the maximum scour point to the furthest downstream and upstream point in the direction of flow, respectively, $a_1$ and $a_2$. In transverse direction to the flow from the maximum scour point towards the furthest point towards are the hydraulic right ($b_1$) and hydraulic left ($b_2$). (**b**) For contour line at the height of 500 mm is the semi-axis in the direction of flow from the maximum scour point to the furthest downstream and upstream point, respectively, $A_1$ and $A_2$; semi-axis length in the transverse direction to the flow from the maximum scour point is towards the furthest point towards the hydraulic right and left, respectively $B_1$ and $B_2$.

Furthermore, while the jet was aligned with the axis of the spillway, positioned in the middle of the experimental channel and the sand uniformly distributed, it was observed that the main axis in the longitudinal direction was deviated. This phenomenon may be due to the presence of small imperfections in the physical model that are difficult to detect, which can cause a certain asymmetry in the real behavior. In addition, the test channel has a large lateral observation window, so the roughness of their lateral walls is different. This introduces a real asymmetry that could be the main cause of the asymmetry observed in some of the tests. For this reason, the presence of the scour hole with longitudinal axes in the direction of the channel, not rectilinear, was resolved by dividing the central axis into two parts: $a_1$ and $a_2$, which are obtained by joining the point of maximum erosion with the level curve of 400 mm in the two furthest points; the same geometric criterion was applied to the transverse axis to obtain $b_1$ and $b_2$. The results obtained with the proposed geometric criterion were compared with the results obtained from applying the morphometric theory of lakes [52] that defines the parameters *Lmax, Bmax, Le,* and *Be.* Figure 15 represents the application of the two criteria to a specific case. The results obtained with both techniques were quite similar. Comparing the values of *a,* obtained from the sum of $a_1$ and $a_2$, with *Lmax,* and the values of *b,* equal to the sum of $b_1$ and $b_2$, with *Bmax,* the MRE was around 2%, so it was finally decided to maintain the geometric criterion here proposed (Figure 16).

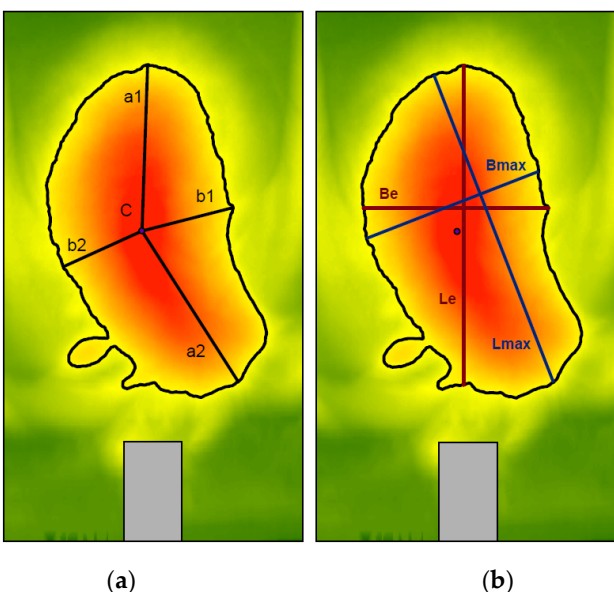

(a)                                                    (b)

**Figure 15.** (**a**) Scour hole axes according to the criterion here proposed. (**b**) According to the morphometric criterion used in lakes: *Lmax* is the line that connects the two points of the lakeshore that are farthest away, *Bmax* is the line perpendicular to *Lmax* that connects the two most distant points of the shore, *Le* is the line that connects the two points of the shore of the lake that are farthest in the direction of the current, *Be* is the line perpendicular to *Le* that connects the two most distant points. In flow direction, the joining of the point of maximum erosion with the contour level of 400 mm in the two furthest points is given by $a_1$ and $a_2$; in cross direction, the joining of the point of maximum erosion with the contour level of 400 mm in the two furthest points is given by $b_1$ and $b_2$.

Once the parameters $a_1$, $a_2$, $b_1$, and $b_2$ were measured, $A_1$, $A_2$, $B_1$, and $B_2$ were determined by linear extrapolation to the 500 mm curve. To verify that the extrapolation was acceptable, we first selected the tests that presented the 500 mm contour closed and determined directly the axes $A_1$, $A_2$, $B_1$, and $B_2$ (Table 6). These experimental values were then compared with those obtained by extrapolation, and the error was calculated. The MRE did not exceed 15.22%. Taking into account the uncertainties of the erosion process, the extrapolation was considered acceptable.

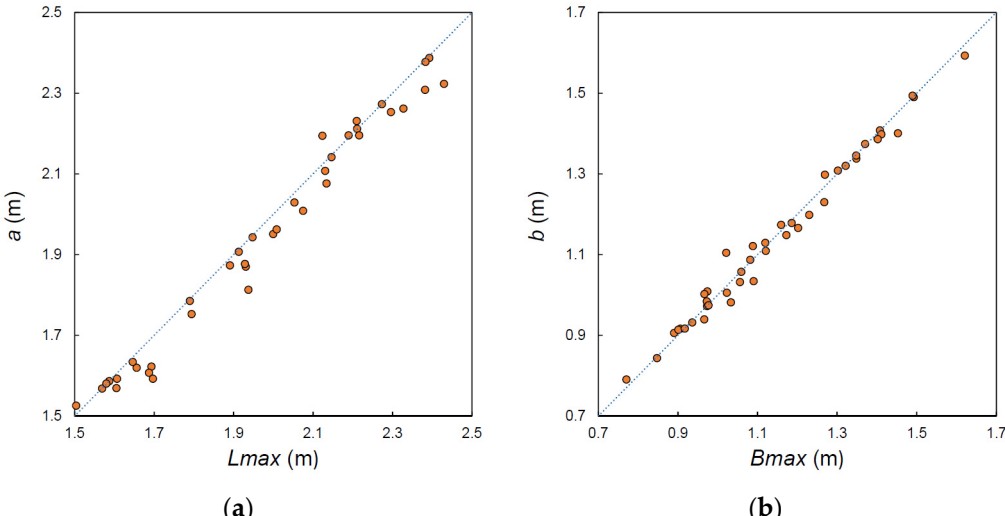

**Figure 16.** (**a**) Comparison of parameters *a* and *Lmax;* (**b**) *b* and *Bmax*.

**Table 6.** Absolute and relative errors between extrapolated and measured values and mean absolute and corresponding relative mean error of: (**a**) axis $A_1$, (**b**) axis $A_2$, (**c**) axis $B_1$, and (**d**) axis $B_2$.

| (a) | | | | | |
|---|---|---|---|---|---|
| **Practice** | $A_{1\ extr}$ | $A_{1\ exp}$ | | **AE (m)** | **RE** |
| A4_h0_17 | 0.891 | 0.895 | | 0.004 | 0.005 |
| A4_h0_18 | 1.035 | 1.027 | | 0.008 | 0.007 |
| A4_h0_18 | 0.983 | 0.974 | | 0.008 | 0.009 |
| A4_h6_17 | 0.873 | 1.085 | | 0.211 | 0.195 |
| A4_h6_18 | 0.991 | 0.916 | | 0.075 | 0.082 |
| A4_h6_20 | 1.027 | 1.035 | | 0.007 | 0.007 |
| B3_h0_20 | 1.158 | 1.210 | | 0.051 | 0.042 |
| B3_h6_17 | 1.026 | 0.982 | | 0.044 | 0.045 |
| B4_h6_18 | 0.989 | 1.052 | | 0.063 | 0.060 |
| B4_h6_20 | 1.062 | 1.095 | | 0.032 | 0.029 |
| C4_h0_17 | 1.032 | 1.001 | | 0.031 | 0.031 |
| C4_h0_20 | 1.058 | 1.120 | | 0.062 | 0.056 |
| C4_h6_17 | 0.920 | 0.878 | | 0.042 | 0.048 |
| C4_h6_20 | 0.901 | 0.931 | | 0.030 | 0.033 |
| | | | MAEv | 0.048 | |
| | | | MREv | | 0.046 |

| (b) | | | | | |
|---|---|---|---|---|---|
| **Practice** | $A_{2\ extr}$ | $A_{2\ exp}$ | | **AE (m)** | **RE** |
| A4_h0_17 | 1.163 | 1.244 | | 0.081 | 0.065 |
| A4_h0_18 | 1.075 | 1.114 | | 0.038 | 0.034 |
| A4_h0_18 | 1.285 | 1.516 | | 0.231 | 0.152 |
| A4_h6_17 | 1.082 | 1.137 | | 0.056 | 0.049 |
| A4_h6_18 | 1.042 | 1.280 | | 0.238 | 0.186 |
| A4_h6_20 | 1.067 | 1.189 | | 0.122 | 0.102 |
| B3_h0_20 | 1.468 | 1.519 | | 0.051 | 0.034 |
| B3_h6_17 | 2.008 | 1.525 | | 0.483 | 0.316 |
| B4_h6_18 | 1.027 | 0.997 | | 0.030 | 0.030 |
| B4_h6_20 | 1.137 | 1.764 | | 0.627 | 0.356 |
| C4_h0_17 | 1.426 | 1.245 | | 0.181 | 0.145 |
| C4_h0_20 | 1.518 | 1.559 | | 0.042 | 0.027 |
| C4_h6_17 | 1.650 | 1.517 | | 0.133 | 0.088 |
| C4_h6_20 | 1.468 | 1.200 | | 0.269 | 0.224 |
| | | | MAEv | 0.184 | |
| | | | MREv | | 0.129 |

**Table 6.** *Cont.*

| (c) | | | | | |
|---|---|---|---|---|---|
| **Practice** | $B_{1\,extr}$ | $B_{1\,exp}$ | | **AE (m)** | **RE** |
| A4_h0_17 | 0.787 | 0.795 | | 0.009 | 0.011 |
| A4_h0_18 | 0.810 | 0.830 | | 0.020 | 0.024 |
| A4_h0_18 | 0.915 | 0.813 | | 0.102 | 0.125 |
| A4_h6_17 | 0.725 | 0.753 | | 0.027 | 0.036 |
| A4_h6_18 | 0.914 | 0.844 | | 0.070 | 0.083 |
| A4_h6_20 | 0.930 | 0.884 | | 0.046 | 0.052 |
| B3_h0_20 | 0.930 | 1.256 | | 0.326 | 0.259 |
| B3_h6_17 | 0.752 | 0.846 | | 0.094 | 0.111 |
| B4_h6_18 | 0.736 | 0.734 | | 0.003 | 0.004 |
| B4_h6_20 | 0.861 | 0.837 | | 0.023 | 0.028 |
| C4_h0_17 | 1.033 | 1.087 | | 0.054 | 0.050 |
| C4_h0_20 | 0.768 | 0.809 | | 0.041 | 0.051 |
| C4_h6_17 | 0.579 | 0.694 | | 0.115 | 0.166 |
| C4_h6_20 | 0.752 | 1.110 | | 0.358 | 0.322 |
| | | | MAEv | 0.092 | |
| | | | MREv | | 0.094 |

| (d) | | | | | |
|---|---|---|---|---|---|
| **Practice** | $B_{2\,extr}$ | $B_{2\,exp}$ | | **AE (m)** | **RE** |
| A4_h0_17 | 0.900 | 0.979 | | 0.079 | 0.080 |
| A4_h0_18 | 0.989 | 1.147 | | 0.159 | 0.139 |
| A4_h0_18 | 0.999 | 1.113 | | 0.114 | 0.103 |
| A4_h6_17 | 0.877 | 1.085 | | 0.208 | 0.191 |
| A4_h6_18 | 0.892 | 1.136 | | 0.244 | 0.215 |
| A4_h6_20 | 0.910 | 0.993 | | 0.083 | 0.083 |
| B3_h0_20 | 0.794 | 0.921 | | 0.127 | 0.138 |
| B3_h6_17 | 0.765 | 0.991 | | 0.226 | 0.228 |
| B4_h6_18 | 0.772 | 1.043 | | 0.271 | 0.260 |
| B4_h6_20 | 0.967 | 1.206 | | 0.239 | 0.198 |
| C4_h0_17 | 0.708 | 0.813 | | 0.105 | 0.129 |
| C4_h0_20 | 1.221 | 1.343 | | 0.123 | 0.091 |
| C4_h6_17 | 0.744 | 0.920 | | 0.175 | 0.191 |
| C4_h6_20 | 0.652 | 0.712 | | 0.060 | 0.084 |
| | | | MAEv | 0.158 | |
| | | | MREv | | 0.152 |

It was observed that the longitudinal axis (*A*) decreases when the flip angle (*α*) increases (Figure 17), in agreement with the fact that erosion develops more vertically for high angles and progresses downstream for lower angles, which result in a longer scour hole.

The values of $A_1$ and $A_2$ were considered to calculate the length of the major axis *A*, and a linear regression type adjustment was performed. Cases where the scour hole reached the foot of the trampoline were excluded, which could distort the result. The calibration and validation sets were differentiated. The formulas obtained for $A_1$ and $A_2$ (Equations (4) and (5)), respectively, present an $R^2$ of 0.984 and 0.980. Considering the validation data, for parameter $A_1$ the MAEv is 0.13 m, and the MREv is 10.70%, and for $A_2$, the MAEv is 0.12 m, and the MREv is 8.40%. The value of *A* (Equation (6)) results from the sum of $A_1$ and $A_2$ and presents a MAE equal to 0.19 m and an MRE of 7.40%.

$$A_1 = \left(0.18\,t + 1.30\;\cos^2\theta\right) \tag{4}$$

$$A_2 = \left(0.85\,t + 2.26\;\cos^2\theta\right) \tag{5}$$

$$A = A_1 + A_2 \tag{6}$$

where $t$ is the scour depth and $\theta$ is the impingement angle.

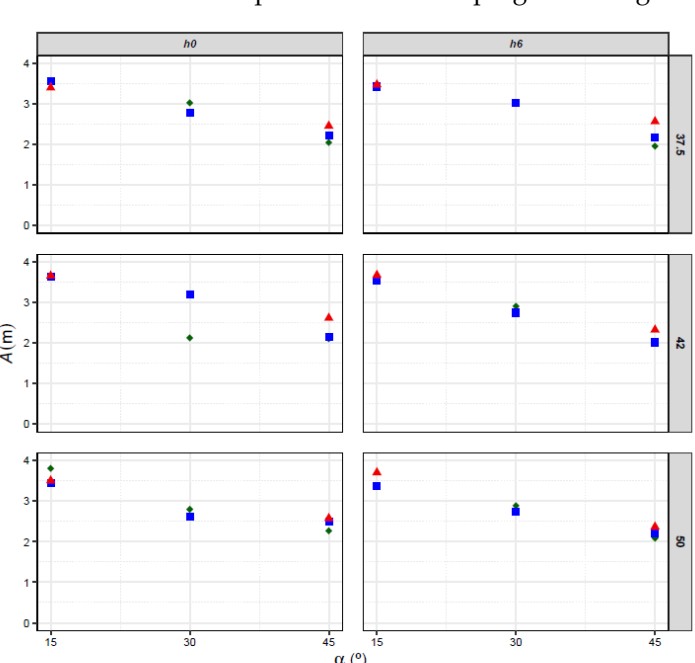

**Figure 17.** Evolution of the maximum length ($A$) of the scour hole with the flip angle ($\alpha$), separating the data by the height water cushion (0.00 m or 0.06 m; columns), by the flow rate (37.5 L/s, 42 L/s, 50 L/s; lines), and by the radius (0.2, 0,3, 0.4 m; shape and color).

It can be observed that Equation (4) has a good correspondence with the experimental values (Figure 18) and that parameter $A_1$ increases when $\cos^2\theta$ increases, confirming that the scour hole lengthens when the impingement angle ($\theta$) decreases. Equation (5) fits the experimental results quite well for cases that have a radius other than 0.40 m (Figure 19).

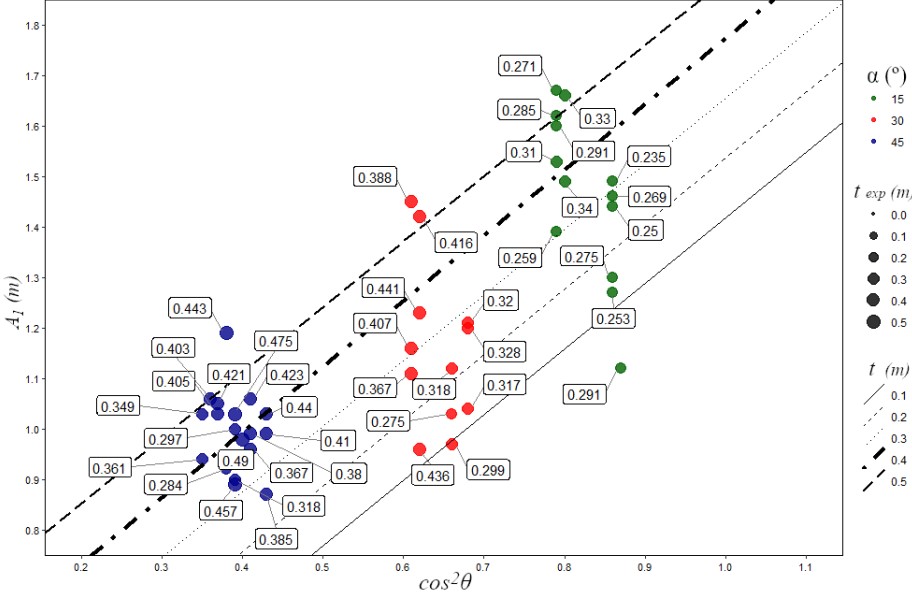

**Figure 18.** Lines obtained with Equation (4) for various values of scour depth ($t$), covering the range of experimental values, including in the graph ($A_1$; $\cos^2\theta$) the experimental points with the corresponding value of $t$.

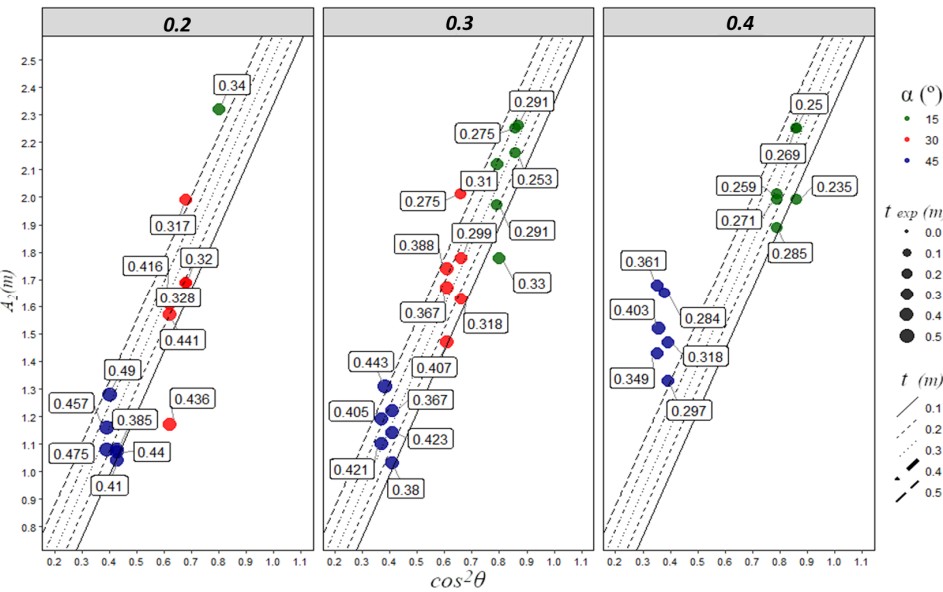

**Figure 19.** Lines obtained with Equation (4) for various values of scour depth ($t$), covering the range of experimental values, including in the graph ($A_2$; $cos^2\theta$) the experimental points with the corresponding value of $t$ and separating the data as a function of the radius $R$.

### 3.4. Erosion Basin Shape

The shape of the limit scour hole was characterized by the relationship between its longitudinal and transverse axes $A/B$, which we call the circularity index. This index would adopt the unit value for a perfect circle, and its value increases as the shape becomes more elongated. Tests have clearly shown that for a high flip angle (45°), the scour hole has a rounded or quasi-circular shape, while it is more elongated for smaller flip angles. Furthermore, it is observed that the circularity index $A/B$ increases when the flip angle ($\alpha$) decreases (Figure 20). When the scour hole reaches the foot of the flip bucket, its shape is affected by this obstacle. Therefore, these cases were not considered for the regression adjustment.

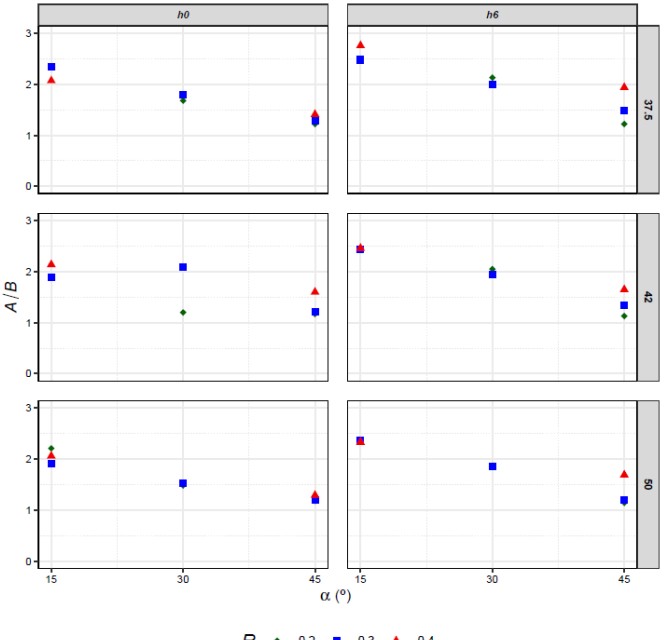

**Figure 20.** Evolution of the circularity index $A/B$ with the flip angle ($\alpha$), separating in two columns the cases without water cushion $h_0$ from those with water cushion $h_6$ and separating in three rows the test flows rates (37.5 L/s, 42 L/s, 50 L/s). Radius ($R$) is indicated by color and shape.

The circularity index can be obtained as *a/b* or *A/B*, since homologous axes are homothetic. For the determination of the formula of the circularity index, a linear regression type adjustment was performed. The formula obtained (Equation (7)) has an $R^2 = 0.985$. Applying the formula to validation cases results in an MAEv of 0.15 m and an MREv of 8.70%.

$$\frac{A}{B} = \left( 0.49 \frac{R}{t} + 1.73 \cos\theta \right) \tag{7}$$

where *t* is the scour depth, *θ* is the impingement angle, and *R* the radius of flip bucket.

It can be observed that Equation (7) has a good correspondence with the experimental values (Figure 21). The parameter *A/B* increases when the cos*θ* increases, as expected due to the fact that the scour hole lengthens when the impingement angle (*θ*) decreases.

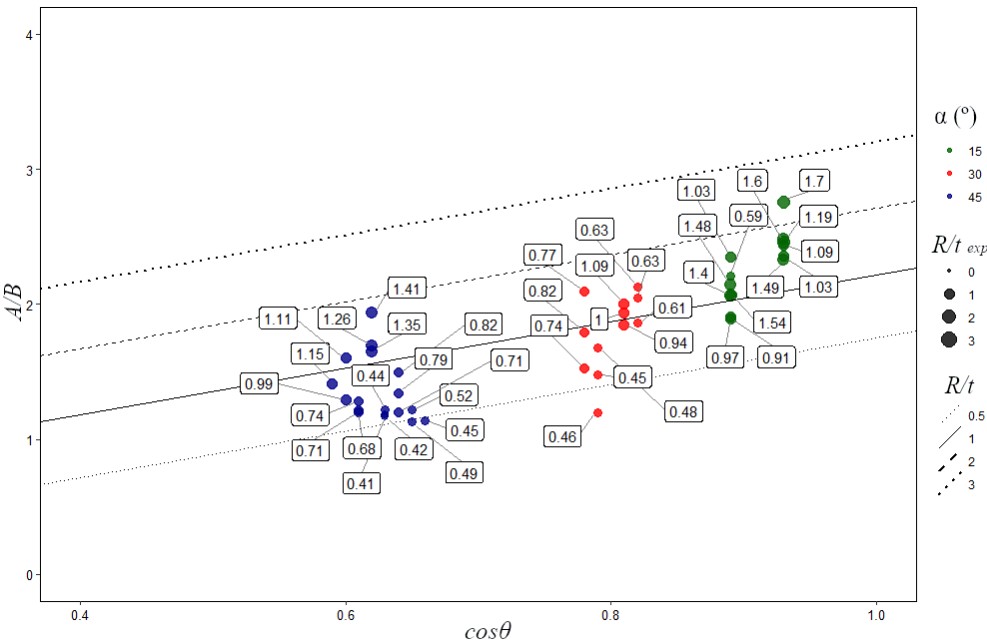

**Figure 21.** Lines obtained with equation for various values of *R/t*, covering the range of experimental values, including in the graph (*A/B*; cos*θ*) the experimental points with the corresponding value of *R/t* and separating the data as a function of the flip angle (*α*).

### 3.5. Procedure for Estimating the Location, the Size, and the Shape of the Erosion Basin

Based on the formulas described above, a simple procedure is proposed to characterize the limit erosion basin's location, shape, and size that takes into account the geometric parameters that define a cylindrical flip bucket (Figure 22). The necessary data are (a) the scour depth (*t*), which the designer can estimate by one of the available formulas, to be chosen by the designer (for example Equation (1)); (b) the radius of flip bucket (*R*); (c) the flip angle (*α*); (d) the vertical distance from the lip of the flip bucket to the level of the reservoir ($z_0$); (e) the vertical distance from the lip of the flip bucket to the surface of the water downstream ($z_1$); and (f) the impingement angle of incidence (*θ*), which can be determined with equations present in the literature [50].

The process has the following steps:

Step 1: Once the scour depth (*t*) is known, we estimate $L_c$, the distance on the horizontal plane from the foot of the flip bucket to point C, where the erosion is maximum. The designer can use Equations (2) or (3) for this.

Step 2: Assuming that two tangent semi-ellipses approximate the shape of the scour hole with a common transverse axis, we proceed by determining the two longitudinal semi-axes $A_1$ and $A_2$ (Equations (4) and (5)). The total length of the longitudinal axis *A* is obtained from the sum of $A_1$ and $A_2$.

Step 3: Check if $L_c > A_2$. If the answer is positive, it should be expected that the limit scour hole would not affect the base of the flip bucket.

Step 4: The circularity index $A/B$ is determined by Equation (7).

Step 5: Once determined, $A$ and the ratio $A/B$, the axis $B$ can immediately be obtained.

In order to validate the procedure, the cases used for the validation of the formulas, not for their calibration, were considered. The limit scour holes resulting from the application of the proposed procedure were drawn and compared with the experimental data (Figure 23). Table 7 summarizes the differences between the experimental and calculated data quantified as mean absolute error (MAEv) and mean relative error (MREv) for all the studied parameters. It can be observed that the MREv is, for all the dimensions that define the limit scour hole in its shape and size, less than 10%, while for the $L_c$ parameter, the MREv is less than 17.5%. It should be noted that this procedure concerns a flip bucket with a certain width, so that width and shape cannot be extrapolated to other cases. However, the estimation of length and position of the limit erosion basin are the most relevant parameters from a design or safety point of view.

A question of great importance is to know if erosion might affect the structure, that is, if the parameter $A_2$ is greater than the parameter $L_c$; in this circumstance, it should be expected that the erosion would affect the flip bucket. Equation (5) (calculation of $A_2$) is applied using the validation tests (Table 8). It is observed that in cases where the 500 mm contour line touches the trampoline (Figure 24), the value of $A_2\_cal$ must be greater than the mean value of $\overline{L_c}$ calculated with Equations (2) and (3); only in two cases (A4_h0_17; B4_h0_18) the value of $A_2\_cal$ is not greater than the mean value of $\overline{L_c}$, although the 500 mm contour line touches the trampoline. In any case, this observation further validates the proposed formula because it is correct ten times out of twelve.

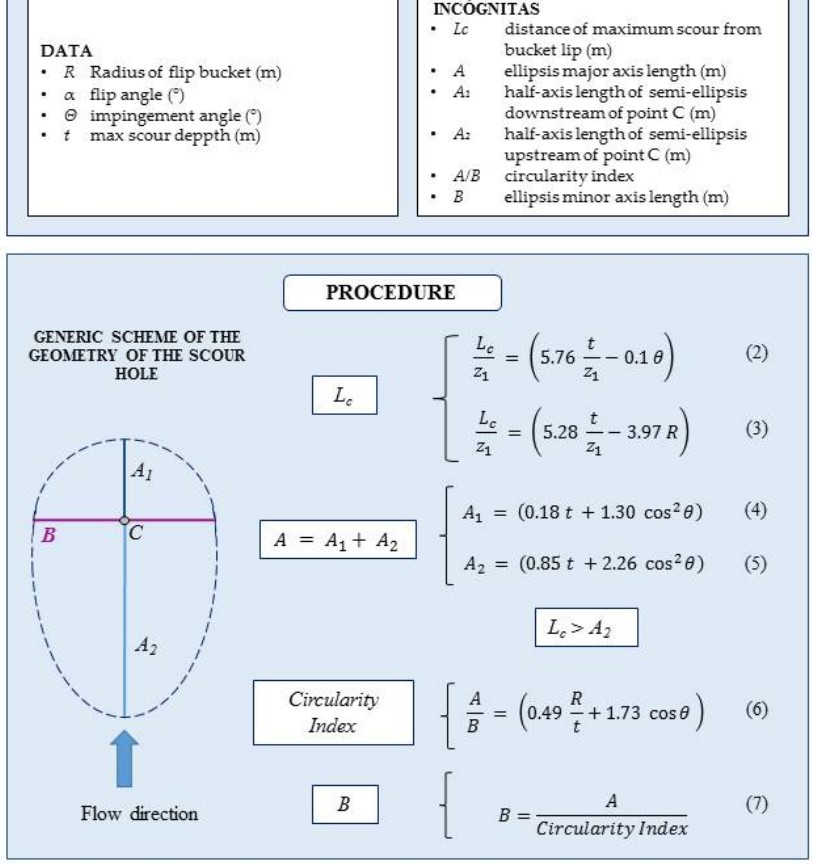

**Figure 22.** Procedure for the estimation of the position, size, and shape of the limit scour hole.

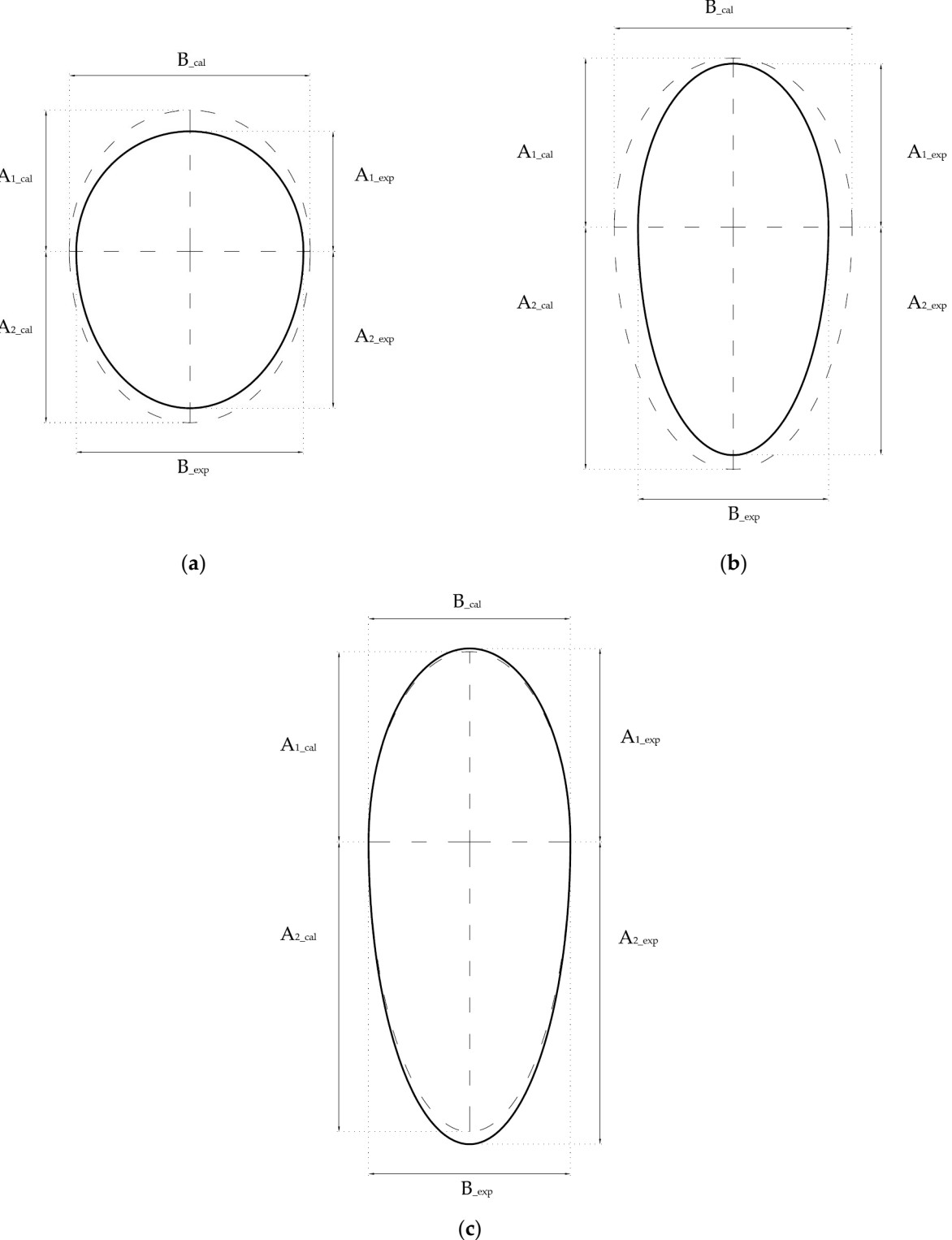

**Figure 23.** Comparison between the shape, size, and position of limit scour hole obtained experimentally (continuous line) and those determined by the proposed procedure (dashed line). Some examples: (**a**) $R = 0.2$ m, $\alpha = 45°$, $Q = 37.5$ L/s and without water cushion. (**b**) $R = 0.2$ m, $\alpha = 30°$, $Q = 50$ L/s and with water cushion of 0.06 m. (**c**) $R = 0.4$ m, $\alpha = 15°$, $Q = 42$ L/s and with water cushion of 0.06 m.

**Table 7.** MAEv and MREv of the calculated parameters of the scour hole taking the measured data as a reference, considering all the validation tests.

| Parameter | $L_c(\theta)$ (m) | $L_c(R)$ (m) | $A_1$ (m) | $A_2$ (m) (m) | $A$(m) (m) | $A/B$ | $B$(m) (m) |
|---|---|---|---|---|---|---|---|
| MAEv (m) | 0.19 | 0.26 | 0.126 | 0.12 | 0.19 | 0.15 | 0.15 |
| MREv (%) | 12.76 | 17.44 | 10.70 | 8.40 | 7.40 | 8.70 | 9.90 |

**Table 8.** Comparison of the value of $A_2$, calculated with Equation (5), with the experimental and calculated values of $L_c$ for validation tests.

| Validation Test | $t$ (m) | $\cos^2\theta$ | $A_{2\_exp}$ (m) | $A_{2\_cal}$ (m) | $L_{c\ exp}$ (m) | $L_c(\theta)$ (m) | $L_c(R)$ (m) | Reached Trampoline | $A_{2\_cal} > \overline{L_c}$ |
|---|---|---|---|---|---|---|---|---|---|
| A3_h0_20 | 0.441 | 0.62 | 1.568 | 1.785 | 1.594 | 2.061 | 2.228 | no | no |
| A3_h6_18 | 0.320 | 0.68 | 1.694 | 1.800 | 1.607 | 1.612 | 1.637 | yes | yes |
| A4_h0_17 | 0.457 | 0.39 | 1.163 | 1.271 | 1.596 | 1.819 | 2.287 | yes | no |
| A4_h6_18 | 0.410 | 0.43 | 1.042 | 1.314 | 1.572 | 1.877 | 2.087 | no | no |
| B2_h0_17 | 0.291 | 0.79 | 1.970 | 2.037 | 1.429 | 1.377 | 1.405 | yes | yes |
| B2_h6_20 | 0.291 | 0.87 | 2.262 | 2.202 | 1.704 | 1.568 | 1.477 | yes | yes |
| B3_h0_17 | 0.367 | 0.61 | 1.666 | 1.681 | 1.641 | 1.569 | 1.771 | yes | = |
| B3_h6_20 | 0.318 | 0.66 | 1.627 | 1.769 | 1.756 | 1.548 | 1.584 | yes | yes |
| B4_h0_18 | 0.421 | 0.37 | 1.098 | 1.202 | 1.514 | 1.443 | 1.999 | yes | no |
| B4_h6_17 | 0.367 | 0.41 | 1.216 | 1.228 | 1.594 | 1.469 | 1.786 | no | no |
| C2_h0_20 | 0.285 | 0.79 | 1.885 | 2.033 | 1.507 | 1.334 | 1.324 | yes | yes |
| C2_h6_18 | 0.250 | 0.86 | 2.246 | 2.154 | 1.497 | 1.322 | 1.235 | yes | yes |

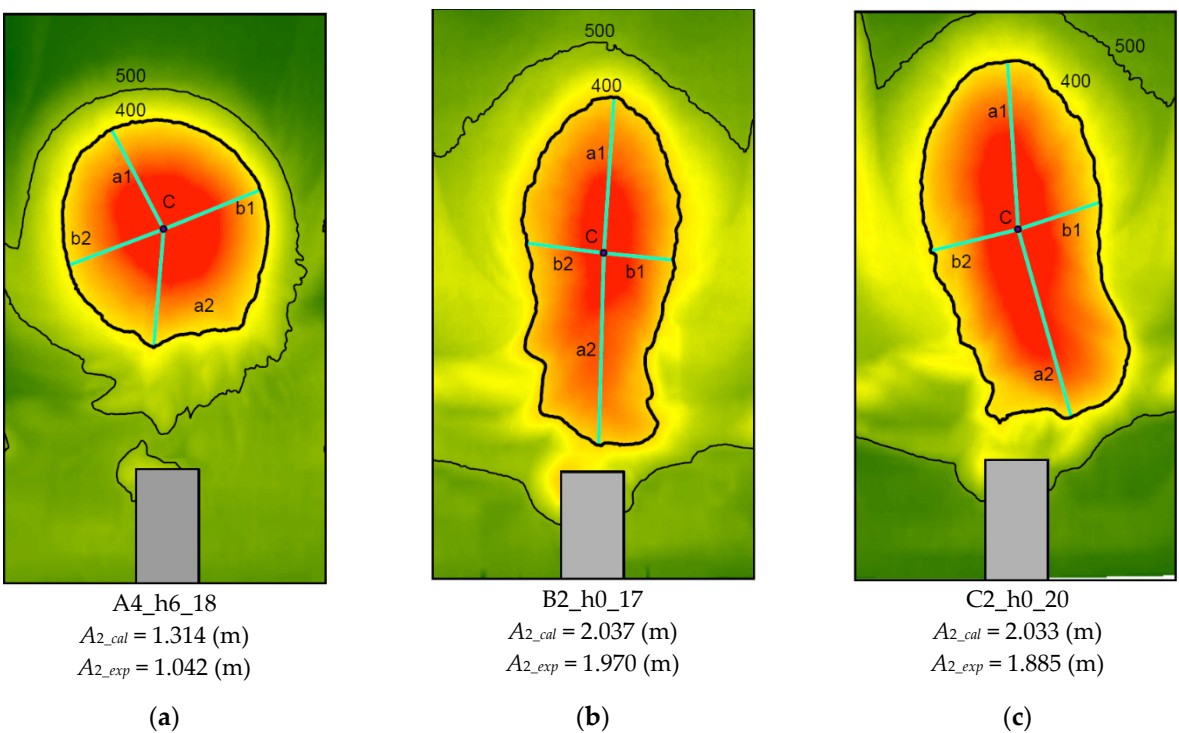

A4_h6_18

$A_{2\_cal}$ = 1.314 (m)

$A_{2\_exp}$ = 1.042 (m)

(**a**)

B2_h0_17

$A_{2\_cal}$ = 2.037 (m)

$A_{2\_exp}$ = 1.970 (m)

(**b**)

C2_h0_20

$A_{2\_cal}$ = 2.033 (m)

$A_{2\_exp}$ = 1.885 (m)

(**c**)

**Figure 24.** Digital terrain model for some tests, with contour lines at 400 mm and 500 mm, and longitudinal ($a_1$ and $a_2$) and transverse axes ($b_1$ and $b_2$) closed on curve line 400 mm, for three tests: (**a**) A4_h6_18 ($R$ = 0.2 m, $\alpha$ = 45°, with a water cushion of 0.06 m and Q = 42.00 L/s), (**b**) B2_h0_17 ($R$ = 0.3 m, $\alpha$ = 15°, without a water cushion and Q = 37.50 L/s), (**c**) C2_h0_20 ($R$ = 0.4 m, $\alpha$ = 15°, without a water cushion and Q = 50.00 L/s). For every test, the calculated and measured value of $A_2$ (the semi-axis length in the direction of flow from the maximum scour point to the furthest upstream point) is provided.

A different shape and orientation of the limit scour hole can be seen in the same Figure 24; in particular, the shape can be more circular or more elliptical, and the orientation can be more leftward or rightward. This can be explained considering the flip angle; in fact, when this angle is small, the horizontal component of the jet velocity is greater than the vertical, causing greater erosion in the longitudinal direction; it means a greater amount of energy is dissipated in direction of the flow. Regarding shape, the dimension in the direction of the flow is the most relevant characteristic as the dimension transversal to the flow depends on the width of the spillway, which remains constant in this study. For the orientation to the left or right, there is not a plausible explanation. An explication of these changes of orientation can be determined by random small imperfections and a not perfect symmetry of the facility given by the different naturalness of the experimental channel walls, which would justify that the orientation does not maintain a clear pattern.

*3.6. Limitations of the Present Study*

This study provides the designer or engineer with one more tool to design or evaluate the safety of a dam; nevertheless, it has some limitations:

- The height of the spillway is constant, so it has not allowed us to obtain a formula in order to estimate the depth of erosion depending on the flow and the geometric characteristics of the spillway; one of the formulas provided by any other author can be used for this purpose;
- The spillway width is constant, so it has not allowed us to provide a direct formula for the estimation of the total width of the limit scour hole;
- A wider test channel would reduce the effect of the walls, which are most likely the reason for the deviations observed in the limit scour hole.

Depending on the particular characteristics of the case, the preliminary or detailed design phase and the safety study, and the available budget, the responsible engineer will judge the tool of this study sufficient or not. For example, if it is an embankment dam founded on alluvial material with a significant thickness, a better approximation can be expected than if the foundation of the dam is made of competent rock.

In any case, this study can be useful in order to make a simple estimation of the limit erosion basin when making decisions about the design itself or about the need of undertaking more detailed work.

**4. Conclusions**

The designer should be aware of the significant influence of the design of the flip bucket on the position, size, and shape of the limit erosion basin.

For the first time, the influence of flip bucket design, flip angle, and radius on the position, dimensions, and shape of the limit erosion basin has been investigated. This study provides some formulas and a procedure to estimate that position, size, and shape of the limit erosion basin. In addition, this work reveals and quantifies the influence of the lip angle on the formation of the limit erosion basin, and it evinces a surprising relationship between the radius of the flip bucket and the depth of the scour hole.

The practical utility of these results is immediate, since the designer can adapt the design of the flip bucket to avoid affecting auxiliary structure, the hillsides, or the spillway and dam itself, proceeding from the safety side as a limit erosion basin is handled.

The flip angle and also the radius chosen in the design stage, together with the flow discharge and configuration of the riverbed, determine the depth, length, width, position, and shape of the limit erosion basin. Establishing a priori the position and size of the scour hole allows to control the potential affection to appurtenant works and the flip bucket itself. The same information is useful for the safety assessment of dams in operation when the limit erosion hole has not yet fully developed.

A methodology is here proposed to estimate the position, size, and shape of the limit scour hole, depending on the geometric characteristics of the flip bucket. Whether or not the estimated size is reached depends on the geological/geotechnical characteristics

of the impact area. The main conclusions of the research can be summarized in the following points:

- Increasing the radius of the flip bucket allows a reduction in the depth of the scour hole. Although the cost of the structure is higher for a greater radius, this extra cost might in some cases be justified for the extra safety level achieved.
- The greater the flip angle is, in the range of tested angles, 15° to 45°, the greater the depth of erosion is.
- Scour hole is longer in the riverbed direction and less deep for low flip angles.
- For the length of flip bucket tested, the scour hole is quasi-circular for a flip angle of 45° and more elongated for lower angles. However, the influence of the lip length is evident, so a different shape should be expected for different lip lengths.
- The plan position of the point where the depth erosion is maximum moves away from the flip bucket with increasing flip angles between 0° and 30°. However, it is nearer the structure for a flip angle of 45°. Two opposite effects might explain this fact: increasing the angle increases the launch scope, up to 45°, but a greater angle of incidence makes the erosion more vertical, so the scour hole develops less in the direction of the riverbed and more in depth.
- Empirical formulas were derived from the experimental data to estimate the position, size and shape of the scour hole. However, it should be noted that a different width, and so shape, should be expected for different lengths of the flip bucket lip, which is a parameter not considered in this experimental research. More tests are needed with different lip lengths.
- A methodology is proposed, using the above mentioned formulas, to estimate position, size, and shape of the scour hole, which was fitted to a combination of two semi-ellipses.
- The proposed methodology was used with success to determine whether the scour hole is likely to affect the flip bucket structure, comparing the length of the scour hole upstream of the point where the depth is maximum with the distance from that point to the flip bucket. If the scour hole overlaps with the structure, affection is likely to occur.

**Author Contributions:** Conceptualization, M.Á.T.; methodology, M.Á.T. and R.P.; validation, R.P.; formal analysis, R.P.; investigation, R.P.; resources, M.Á.T.; data curation, R.P.; writing—original draft preparation, R.P.; writing—review and editing, M.Á.T.; visualization, R.P.; supervision, M.Á.T.; project administration, M.Á.T. All authors have read and agreed to the published version of the manuscript.

**Funding:** This research received no external funding.

**Institutional Review Board Statement:** Not applicable.

**Informed Consent Statement:** Not applicable.

**Data Availability Statement:** The data relating to this work are collected in Raffaella Pellegrino's doctoral thesis available in the library of Universidad Politécnica de Madrid.

**Acknowledgments:** We are grateful to the members of the research group SERPA-Dam Safety Research for the support provided.

**Conflicts of Interest:** The authors declare no conflict of interest.

## Abbreviations

The following abbreviations are used in this paper:

| | |
|---|---|
| $A$ | total length of the limit scour hole in the river longitudinal direction |
| $A/B$ | circularity index |
| $A_1$ | semi-axis length in the direction of flow from the maximum scour point to the furthest downstream point |

| $A_2$ | semi-axis length in the direction of flow from the maximum scour point to the furthest upstream point |
| --- | --- |
| $AR$ | absolute error |
| $B$ | total width of the limit scour hole |
| $B_1$ | semi-axis length in the transverse direction to the flow from the maximum scour point towards the furthest point towards the hydraulic right |
| $B_2$ | semi-axis length in the transverse direction to the flow from the maximum scour point towards the furthest point towards the hydraulic left |
| $D$ | total scour depth |
| $D_{exp}$ | experimental total scour depth |
| $D_{cal}$ | calculated total scour depth |
| $d_{50}$ | grain diameter at 50% of weight |
| $d_{90}$ | grain diameter at 90% of weight |
| $H$ | total head (distance between upstream and downstream water level) |
| $h_2$ | tailwater depth (downstream water level) |
| $L_c$ | distance of maximum scour from bucket lip |
| $MAE$ | mean absolute error |
| $MRE$ | mean relative error |
| $Q$ | flow rate |
| $q$ | unit flow rate |
| $R$ | radius of flip bucket |
| $RE$ | relative error |
| $T$ | scour depth |
| $z_1$ | distance from the flip bucket's lip to the downstream water level |
| $z_o$ | distance between flip bucket's lip and upstream water level |
| $z_p$ | distance from the flip bucket's lip to the ground |
| $\alpha$ | flip angle |
| $\theta$ | impingement jet angle |

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
