# Peer review of "Characterization of the Erosion Basin Shaped by the Jet Flow of Sky-Jump Spillways"

_water, doi:10.3390/w15162930_

Round 1

Reviewer 1 Report

In this paper, the Authors presents the results from series a laboratory experiments on the features of the erosion caused by the jet from a sky-jump spillway, and then propose a procedure to estimate the main characteristics of the erosion in the receiving basin. The topic, even if not groundbreaking, is of interest for Water audience, in particular because many interesting experimental data are presented that could be of interest for future investigations and applications and, moreover, the accuracy of the main formulas employed to estimate the erosion is evaluated. The paper is mainly properly organized (just some repetitions all along the paper could be removed) and the English is fine. Anyway, as a reviewer I feel that some improvements are required to improve the paper impact. General comments are given below.

GENERAL COMMENTS

1.       A discussion on the dependence of the erosion features on the terrain characteristic is missing and should be added as, of course, the erosion in sand should be different from the erosion in rocks. This is valid both in the introduction and in the Physical modeling section, as I have found not any discussion about the sand choice in the laboratory model and its relation with a real terrain at full scale.

2.       What are the highlighted data (in yellow and pale blue) in some tables? Please explain, both ih the main text and in the related captions.

3.       Why are the main (longest) axis of the scour hole not aligned to the spillway? Was the jet not aligned to the spillway axis? Or the sand is not uniformly distributed?

4.       Which are the main experimental parameters in the figures with the scour holes (like 14, 15 and 24)? Do the Authors have any explanation about the different shape (more circular or more elliptical) and orientation (more leftward or rightward) of the scour holes presented?

5.       The limitations of the present study are not properly highlighted, so please add a paragraph about that.

In my opinion, this paper could be of interest for the audience of Water but I believe some relevant improvements are needed before it can be published. I propose the paper for major revisions.

Reviewer 2 Report

The article is devoted to the solution of a technical problem - the assessment of the erosion pit as a result of the flow of costs into the downstream. Various methods of calculation and results of experiments in a model setup are generalized. I would like to see the scientific component of this project. What new results have been obtained? How can these results be applied to real structures? How to take into account the heterogeneous properties of soils and non-stationary modes of operation of the spillway? How to assess the risks of destruction of a structure as a result of the formation of an erosion pit?

Round 2

Reviewer 1 Report

My points have been properly addressed and, in my opinion, the paper is now ready to be published in Water journal.